# Context Mover's Distance & Barycenters: Optimal transport of contexts for building representations

## Abstract

We propose a unified framework for building unsupervised representations of entities and their compositions, by viewing each entity as a histogram (or distribution) over its contexts. This enables us to take advantage of *optimal transport* and construct representations that effectively harness the geometry of the underlying space containing the contexts. Our method captures uncertainty via modelling the entities as distributions and simultaneously provides interpretability with the optimal transport map, hence giving a novel perspective for building rich and powerful feature representations. As a guiding example, we formulate unsupervised representations for text, and demonstrate it on tasks such as sentence similarity and word entailment detection. Empirical results show strong advantages gained through the proposed framework. This approach can potentially be used for any unsupervised or supervised problem (on text or other modalities) with a co-occurrence structure, such as any sequence data. The key tools at the core of this framework are Wasserstein distances and Wasserstein barycenters, hence raising the question from our title.

## 1 Introduction

One of the main driving factors behind the recent successes in machine learning and natural language processing has been the development of better representation methods for data modalities. Examples include continuous vector representations for language (Mikolov et al., 2013; Pennington et al., 2014), Convolutional Neural Network (CNN) based text representations (Collobert & Weston, 2008; Kim, 2014; Kalchbrenner et al., 2014; Severyn & Moschitti, 2015; Deriu et al., 2017), or via other neural architectures such as RNNs, LSTMs (Hochreiter & Schmidhuber, 1997; Kiros et al., 2015), all sharing one central idea – to map input entities to dense vector embeddings lying in a low-dimensional latent space where the semantics of the inputs are preserved.

While these existing methods represent each entity of interest (e.g., a word) directly as a single point in space (i.e., its embedding vector), we here propose a fundamentally different approach. We focus on co-occurrence information of the entities and their contexts (e.g. context words), and leverage embeddings of the contexts instead of the original entities. So instead of a single point per entity, we represent each entity by the *histogram of its contexts*, where the contexts themselves are represented as points in a suitable metric space. This allows us to cast the distance between histograms associated with the contexts as an instance of the *optimal transport problem* (Monge, 1781; Kantorovich, 1942; Villani, 2008).

Our resulting framework then intuitively seeks to minimize the cost of moving the contexts of a given entity to the contexts of another, which motivates the naming *Context Mover's Distance* (CMD). Note that the contexts here can be words, phrases, sentences, or any generic entities co-occurring with our entities to be represented, and these entities further could be of various kinds, including e.g., products such as movies or web-advertisements (Grbovic et al., 2015), nodes in a graph (Grover & Leskovec, 2016), sequence data, or any other entities (Wu et al., 2017). Any co-occurrence structure will allow construction of the histogram information, which is the crucial building block of our approach.

The main motivation for our proposed approach here comes from the domain of natural language, where the entities (words, phrases, or sentences) generally have different semantics depending on the context under which they are present. Hence, it is important that we consider representations

that are able to effectively capture such inherent uncertainty and polysemy, and we will argue that histograms (or probability distributions) over embeddings allows to capture more of this information compared to point-wise embeddings alone. We will call this histogram over contexts embeddings as the *distributional estimate* of our entity of interest, while we refer to the individual embeddings of contexts as *point estimates*.

The connection to optimal transport at the level of entities and contexts paves the way to make better use of its vast toolkit (like Wasserstein distances, barycenters, barycentric coordinates, etc.) for applications in NLP, which in the past has primarily been restricted to document distances of original words (Kusner et al., 2015; Huang et al., 2016), as opposed to contexts. Thanks to the entropic regularization introduced by Cuturi (2013), optimal transport computations can be carried out efficiently in a parallel and batched manner on GPUs.

**Contributions:**

- Employing the notion of optimal transport of contexts as a distance measure, we illustrate how our framework can be of benefit for various important tasks, including word and sentence representations, sentence similarity, as well as hypernymy (entailment) detection. The method is static and does not require any additional learning, and can be readily used on top of existing embedding methods.

- The resulting representations via the transport map give a clear interpretation of the resulting distance (see also Figure 1), on top of the co-occurrence information.

- Our Context Mover's Distance can be used to measure any kind of distance (even asymmetric) between words, by defining a suitable underlying cost on the movement of contexts, which we show can lead to a state-of-the-art metric for word entailment.

- Defining the transport over contexts has the added benefit that the representations are compositional - they directly extend from entities to groups of entities (of any size), such as from word to sentence representations. To this end, we utilize the notion of Wasserstein barycenters, which to the best of our knowledge has never been considered in the past.

The proposed framework is not specific to words or sentences but can allow building unsupervised representations for any entity and composition of entities, where a co-occurrence structure can be devised between entities and their contexts, and we leave this direction for a future work.

## 2  RELATED WORK

Most of the previous work in building representations for natural language has been focused towards vector space models, in particular, popularized through the groundbreaking work in Word2vec (Mikolov et al., 2013) and GloVe (Pennington et al., 2014). The key idea in these models has been to map words which are similar in meaning to nearby points in a latent space. Based on which, many works (Levy & Goldberg, 2014a; Melamud et al., 2015; Bojanowski et al., 2016) have suggested specializing the embeddings to capture some particular information required for the task at hand. One of the problems that still persists is the inability to capture, within just a point embedding, the various semantics and uncertainties associated with the occurrence of a particular word (Huang et al., 2012; Guo et al., 2014).

A recent line of work has proposed the view to represent words with Gaussian distributions or mixtures of Gaussian distributions (Vilnis & McCallum, 2014; Athiwaratkun & Wilson, 2017), or hyperbolic cones (Ganea et al., 2018) for this purpose. Also, concurrent works by Muzellec & Cuturi (2018) and Sun et al. (2018) have suggested using elliptical and Gaussian distributions endowed with a Wasserstein metric respectively. While these already provide richer information than typical vector embeddings, their form restricts what could be gained by allowing for arbitrary distributions. In addition, hyperbolic embeddings (Nickel & Kiela, 2017; Ganea et al., 2018) are so far restricted to supervised tasks (and even elliptical embeddings (Muzellec & Cuturi, 2018) to the most extent), not allowing unsupervised representation learning as in the focus of the paper here. To this end, we propose to represent each word with a distributional estimate (i.e., histogram over context embeddings), that inherently relies upon the empirically obtained co-occurrence information of a word and its contexts. Hence, this naturally allows for the use of optimal transport (or Wasserstein

metric) in the space containing the contexts, and leads to an interpretation (Figure 1) which is not available in the above approaches.

Amongst the few explorations of optimal transport in NLP, i.e., document distances (Kusner et al., 2015; Huang et al., 2016), topic modelling (Rolet et al., 2016; Xu et al., 2018), document clustering (Ye et al., 2017), bilingual lexicon induction (Zhang et al., 2017), or orthogonal Procrustes mapping (Grave et al., 2018), the focus has been on transporting words directly. For example, the Word Mover's Distance (Kusner et al., 2015) casts finding the distance between documents as an optimal transport problem between their bag of words representation. Our approach is different as we consider the transport over contexts instead, and use it to propose a representation for words or entities. This provides the added flexibility to establish any kind of distance between entities and extend the representation to composition of entities in a principled manner, as will be illustrated further through the examples of entailment detection and sentence representation respectively.

## 3 BACKGROUND ON OPTIMAL TRANSPORT

Optimal Transport (OT) provides a way to compare two probability distributions defined over a space $\mathcal{G}$ (commonly known as the ground space), given an underlying distance or more generally a cost of moving one point to another in the ground space. In other terms, it lifts a distance between points to a distance between distributions. Other methods of comparing distributions, such as Kullback-Liebler (KL), squared Hellinger, etc., only focus on the probability mass values, thus ignoring the geometry of the ground space: something which we utilize throughout this work via OT. Also, some divergences like KL are not defined when the supports of distributions under comparison don't match. Hence, we give a short yet formal background on OT in the discrete case.

**Linear Program Formulation.**  Consider an empirical probability measure of the form $\mu = \sum_{i=1}^{n} a_i \delta(x_i)$ where $X = (x_1, \ldots, x_n) \in \mathcal{G}^n$, $\delta(x)$ denotes the Dirac (unit mass) distribution at point $x \in \mathcal{G}$, and $(a_1, \ldots, a_n)$ lives in the probability simplex $\Sigma_n := \left\{ p \in \mathbb{R}_+^n \mid \sum_{i=1}^{n} p_i = 1 \right\}$. Now given a second empirical measure, $\nu = \sum_{j=1}^{m} b_j \delta(y_j)$, with $Y = (y_1, \ldots, y_m) \in \mathcal{G}^m$, and $(b_1, \ldots, b_m) \in \Sigma_m$, and if the ground cost of moving from point $x_i$ to $y_j$ is denoted by $M_{ij}$, then the Optimal Transport distance between $\mu$ and $\nu$ is the solution to the following linear program.

$$\text{OT}(\mu, \nu; M) := \min_{T \in \mathbb{R}_+^{n \times m}} \sum_{ij} T_{ij} M_{ij} \text{ such that } \forall i, \sum_{j} T_{ij} = a_i, \quad \forall j, \sum_{i} T_{ij} = b_j. \quad (1)$$

Here, the optimal $T \in \mathbb{R}^{n \times m}$ is referred to as the *transportation matrix*: $T_{ij}$ denotes the optimal amount of mass to move from point $x_i$ to point $y_i$. Intuitively, OT is concerned with the problem of moving goods from factories to shops in such a way that all the demands are satisfied and the overall transportation cost is minimal.

**Distance.**  When $\mathcal{G} = \mathbb{R}^d$ and the cost is defined with respect to a metric $D_{\mathcal{G}}$ over $\mathcal{G}$ (i.e., $M_{ij} = D_{\mathcal{G}}(x_i, y_j)^p$ for any $i, j$), OT defines a distance between empirical probability distributions. This is the $p$-Wasserstein distance, defined as $W_p(\mu, \nu) := \text{OT}(\mu, \nu; D_{\mathcal{G}}^p)^{1/p}$. In most cases, we are only concerned with the case where $p = 1$ or $2$.

The cost of exactly solving OT problem scales at least in $\mathcal{O}(n^3 \log(n))$ ($n$ being the cardinality of the support of the empirical measure) when using network simplex or interior point methods. Following Cuturi (2013), we consider the entropy regularized Wasserstein distance, $W_{p,\lambda}(\mu, \nu)$, where the search space for the optimal $T$ is instead restricted to a smooth solution close to the extreme points of this linear program. The regularized problem can then be solved efficiently using Sinkhorn iterations (Sinkhorn, 1964), albeit at the cost of some approximation error. The regularization strength $\lambda \geq 0$ controls the accuracy of approximation and recovers the true OT for $\lambda = 0$. The cost of the Sinkhorn algorithm is only quadratic in $n$ at each iteration. Recently, Altschuler et al. (2017) have shown that the Sinkhorn algorithm converges in a number of iterations independent of $n$, thus resulting in an overall complexity of $\widetilde{O}(n^2/\epsilon^3)$ for an $\epsilon$-accurate solution.

**Barycenters.**  Further on in our discussion, we will make use of the notion of averaging in the Wasserstein space. More precisely, the Wasserstein barycenter, introduced by Agueh & Carlier (2011),

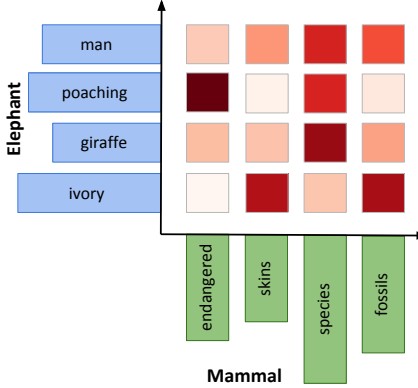

Figure 1: *Illustration of Context Mover's Distance (CMD) (Eq. 3) between elephant & mammal* (when represented with their distributional estimates and using entailment ground metric discussed in Section 7). Here, we pick four contexts at random from a list of top 20 contexts (in terms of PPMI) for the two histograms. Then we plot the transportation matrix (or transport map) $T$ obtained in the process of computing CMD. Note how 'ivory' adjusts its movement towards 'skin' (as in skin color) to allow 'poaching' to be easily moved to 'endangered' as going to other contexts of 'mammal' is costlier for 'poaching', thus focussing on an overall transport cost minimization.

is a probability measure that minimizes the sum of ($p$-th power) Wasserstein distances to the given measures. Formally, given $N$ measures $\{\nu_1, \ldots, \nu_N\}$ with corresponding weights $\eta = \{\eta_1, \ldots, \eta_N\} \in \Sigma_N$, the Wasserstein barycenter can be written as $B_p(\nu_1, \ldots, \nu_N) = \arg\min_\mu \sum_{i=1}^N \eta_i W_p(\mu, \nu_i)^p$. We similarly consider the regularized barycenter $B_{p,\lambda}$, using entropy regularized Wasserstein distances $W_{p,\lambda}$ in the above minimization problem, following Cuturi & Doucet (2014). Employing the method of iterative Bregman projections (Benamou et al., 2015), we obtain an approximation of the solution at a reasonable computational cost.

## 4 METHODOLOGY

In this section, we define the distributional estimate that we use to represent each entity. In view of the guiding example of building text representations, consider each entity to be a word for simplicity.

**Distributional Estimate ($\mathbb{P}_V^w$).**   For a word $w$, its distributional estimate is built from a histogram over the set of contexts $\mathcal{C}$, and an embedding of these contexts into a space $\mathcal{G}$. The histogram essentially measures how likely it is for a word $w$ to occur in a particular context $c$, i.e., probability $p(w|c)$. The exact formulation of this distribution is generally intractable and hence it's common to empirically estimate this by the number of occurrences of the word $w$ in context $c$, relative to the total frequency of context $c$ in the corpus.

Thus one way to build this histogram is to maintain a co-occurrence matrix between words in our vocabulary and all possible contexts, where each entry indicates how often a word and context occur in an interval (or window) of a fixed size $L$. Then, the bin values $(\mathrm{H}^w)_{c \in \mathcal{C}}$ of the histogram $(\mathrm{H}^w)$ for a word $w$, can be viewed as the row corresponding to $w$ in this co-occurrence matrix. In Section 5, we discuss possible modifications of the co-occurrence matrix to improve associations and how to reduce the number of bins in the histogram.

The simplest embedding of contexts is into the space of one-hot vectors of all the possible contexts. However, this induces a lot of sparsity/redundancy in the representation and the distance between such emebddings of contexts does not reflect their semantics. A classical solution would be to instead find a dense low-dimensional embedding of contexts that captures the semantics, possibly using techniques such as SVD or deep neural networks. We denote by $V = (\mathbf{v}_c)_{c \in \mathcal{C}}$ an embedding of the contexts into this low-dimensional space $\mathcal{G} \subset \mathbb{R}^d$, which we refer to as the *ground space*. (We will consider example cases of how this metric can be obtained in Sections 6 and 7.)

Combining the histogram $\mathrm{H}^w$ and the embedding $V$, we represent the word $w$ by the following empirical distribution:

$$\mathbb{P}_V^w := \sum_{c \in \mathcal{C}} (\mathrm{H}^w)_c \, \delta(\mathbf{v_c}). \tag{2}$$

Recall that $\delta(\mathbf{v_c})$ denotes the Dirac measure at the position $\mathbf{v_c}$ of the context $c$. We refer to this representation (Eq. 2) as the *distributional estimate* of the word.

**Distance.**   If we equip the ground space $\mathcal{G}$ with a meaningful metric $D_\mathcal{G}$, then we can subsequently define a distance between the representations of two words $w_i$ and $w_j$, as the solution to the following

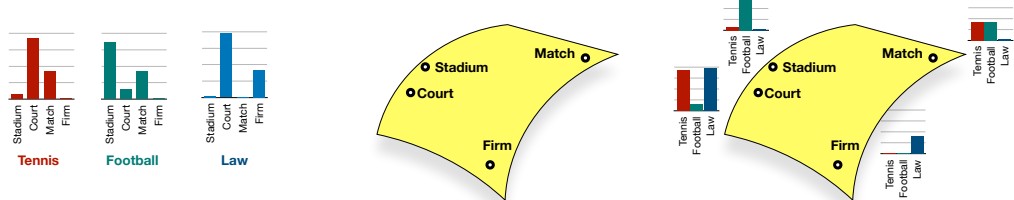

Figure 2: *Illustration of three words, each with their histograms* (left)*, as well as the point estimates of the relevant contexts* (middle)*, and then jointly as their distributional estimates* (right)*. The right figure shows how the support (i.e. context words) of histograms gets assigned on the manifold. For example, the red bars are still associated to the histogram of 'Tennis', but are located at the position of the context vectors of 'Tennis' in the ground space.*

optimal transport problem:

$$\text{CMD}(w_i, w_j; D_{\mathcal{G}}^p) := \text{OT}(\mathbb{P}_V^{w_i}, \mathbb{P}_V^{w_j}; D_{\mathcal{G}}^p) \simeq W_{p,\lambda}(\mathbb{P}_V^{w_i}, \mathbb{P}_V^{w_j})^p. \tag{3}$$

Under this formulation, we call the above distance (Eq. 3) the *Context Mover's Distance* (CMD), borrowing the name from Rubner et al. (2000)'s famous Earth Mover's Distance in computer vision.

**Intuition.**  Two words are similar in meaning if the contexts of one word can be easily transported to the contexts of the other, with this cost of transportation being measured by $D_{\mathcal{G}}$. This idea still remains in line with the distributional hypothesis (Harris, 1954; Rubenstein & Goodenough, 1965) that words in similar contexts have similar meanings, but provides a precise way to quantify it.

**Interpretation.**  In fact, both elements of the distributional estimate: the histogram and point estimates are closely tied together and required to serve as an effective representation. For instance, let's take a toy example and discuss a scenario that might arise when we only have the histogram information. Consider three words, *'Tennis'*, *'Football'*, and *'Law'*, admitting as contexts {*Stadium, Court, Match, Firm*}, with respective histograms shown in left part of Figure 2. Now, if we only took into account the histograms, we would reach the inaccurate conclusion that 'Tennis' is closer in semantics to 'Law' than to 'Football', as there is a considerable overlap at the important context of 'Court' for 'Tennis' and 'Law'. Whereas, together with the point estimate information, it is apparent that the context 'Stadium' (in $\text{H}^{Football}$) can be more cheaply moved to 'Court' (in $\text{H}^{Tennis}$), but moving 'Firm' (in $\text{H}^{Law}$) to some context in $\text{H}^{Tennis}$ is more costly. Lastly, in the reverse scenario of only considering the point estimates, we would lose much of the uncertainty associated about the contexts in which the words occur. We illustrate these scenarios in Figure 2.

**Roadmap.**  In the next section we discuss concretely how this can be applied and for the sake of brevity we restrict to the particular case where contexts consist of single words. Section 6 details how this framework can be extended to obtain representation for a composition of entities via Wasserstein barycenter. Lastly in section 7, we utilize the fact that the CMD in 3 is parameterized by ground cost, and show how this flexibility can be used to define an asymmetric cost measuring entailment.

## 5 Concrete Framework

**Making associations better.**  We consider co-occurrences of a word and a context word if the latter appears in a symmetric window of size $L$ around the target word (the word whose distributional estimate we seek). While each entry of the co-occurrence matrix reflects the co-occurrence count of a target word and its context, the counts alone may not necessarily suggest a strong association between the two. The well-known Positive Pointwise Mutual Information (PPMI) matrix (Church & Hanks, 1990; Levy et al., 2015) addresses this shortcoming, and is defined as follows: $\text{PPMI}(w, c) := \max(\log(\frac{p(w,c)}{p(w) \times p(c)}), 0)$. The PPMI entries are non-zero when the joint probability of co-occurring target and context words is higher than the probability when they are independent. Typically, these probabilities are estimated from the co-occurrence counts in the corpus. Further improvements to the PPMI matrix have been suggested, like in Levy & Goldberg (2014b), and following them we make use of a shifted and smoothed PPMI matrix, denoted by $\text{SPPMI}_{\alpha,s}$ where $\alpha$ and $s$ denote the smoothing and k-shift parameters [1]. Overall, these variants of PPMI enable us to extract better

---

[1]Please refer to Appendix A.2 for the definition of SPPMI and more details such as our column normalization.

semantic associations from the co-occurrence matrix. Hence, the bin values (at context $c$) for the histogram of word $w$ in Eq. 2 can be formulated as: $(\mathbf{H}^w)_c := \frac{\text{SPPMI}_{\alpha,s}(w,c)}{\sum_{c \in \mathcal{C}} \text{SPPMI}_{\alpha,s}(w,c)}$.

**Computational considerations.** The view of optimal transport between histograms of contexts introduced in Eq. 3 offers a pleasing interpretation (see Figure 1). However, it might be computationally intractable in its current formulation, since the number of possible contexts can be as large as the size of vocabulary (if the contexts are just single words) or even exponential (if contexts are considered to be phrases, sentences and otherwise). This is problematic because the Sinkhorn algorithm for regularized optimal transport (Cuturi, 2013, see Section 3) scales roughly quadratically in the histogram size, and the ground cost matrix can also become prohibitive to store in memory. One possible fix is to instead consider a set of representative contexts in this ground space, for example via clustering. We believe that with dense low-dimensional embeddings and a meaningful metric between them, we may not require as many contexts as needed before. For instance, this can be achieved by clustering the contexts with respect to metric $D_{\mathcal{G}}$. Apart from the computational gain, the clustering will lead to transport between more abstract contexts. This will although come at the loss of some interpretability.

Now, consider that we have obtained $K$ representative contexts, each covering some part $\mathcal{C}_k$ of the set of contexts $\mathcal{C}$. The histogram for word $w$ with respect to these contexts can then be written as $\tilde{\mathbb{P}}^w_{\tilde{V}} = \sum_{k=1}^K (\tilde{\mathbf{H}}^w)_k \, \delta(\tilde{\mathbf{v}}_k)$. Here $\tilde{\mathbf{v}}_k \in \tilde{V}$ is the point estimate of the $k^{th}$ representative context, and $(\tilde{\mathbf{H}}^w)_k$ denote the new histogram bin values with respect to the part $\mathcal{C}_k$,

$$(\tilde{\mathbf{H}}^w)_k := \frac{\text{SPPMI}_{\alpha,s}(w,\mathcal{C}_k)}{\sum_{k=1}^K \text{SPPMI}_{\alpha,s}(w,\mathcal{C}_k)}, \text{with} \quad \text{SPPMI}_{\alpha,s}(w,\mathcal{C}_k) := \sum_{c \in \mathcal{C}_k} \text{SPPMI}_{\alpha,s}(w,c). \quad (4)$$

**Summary.** With the above aspects in account and using batched implementations on (Nvidia TitanX) GPUs, it is possible to compute around **13,700** Wasserstein-distances/second (for histogram of size 100). Same also holds for barycenters, where we can compute **4,600** barycenters/second for sentences of length 25 and histogram size of 100. Building this histogram information comes almost for free during the typical learning of embeddings, as in GloVe (Pennington et al., 2014). A practical take-home message of this work thus is to *not throw away the co-occurrence information* e.g. when using GloVe, but to *instead pass it on to our method*.

# 6 SENTENCE REPRESENTATION WITH COMB

Traditionally, the goal of this task is to develop a representation for sentences, that captures the semantics conveyed by it. Most unsupervised representations proposed in the past rely on the composition of vector embeddings for the words, through either additive, multiplicative, or other ways (Mitchell & Lapata, 2008; Arora et al., 2017; Pagliardini et al., 2017). We propose to represent sentences as probability distributions to better capture the inherent uncertainty and polysemy.

Our belief is that a sentence representation is meaningful if it best captures the simultaneous occurrence of the words in it. We hypothesize that a sentence, $S = (w_1, w_2, \ldots, w_N)$, can be efficiently represented via the Wasserstein barycenter (see Section 3) of the distributional estimates of its words,

$$\tilde{\mathbb{P}}_S := B_{p,\lambda}\left(\tilde{\mathbb{P}}^{w_1}_V, \tilde{\mathbb{P}}^{w_2}_V, \ldots, \tilde{\mathbb{P}}^{w_N}_V\right), \quad (5)$$

which is itself again a distribution over the ground space $\mathcal{G}$. We refer to this representation as the *Context Mover's Barycenters* (CoMB) henceforth. Interestingly, the classical weighted averaging of point-estimates, like Smooth Inverse Frequency (SIF) in (Arora et al., 2017) (without principal component removal), can be seen as a special case of CoMB, when the distribution associated to a word is just a Dirac at its point estimate. It becomes apparent that having a rich distributional estimate for a word could turn out to be advantageous.

Since with barycenter representation as in Eq. 5, each sentence is also a distribution over contexts, we can utilize the Context Mover's Distance (CMD) defined in Eq. 3 to define the distance between two sentences $S_1$ and $S_2$, under a given ground metric $D_{\mathcal{G}}$ as follows,

$$\text{CMD}(S_1, S_2; D_{\mathcal{G}}^p) := \text{OT}(\tilde{\mathbb{P}}^{S_1}_V, \tilde{\mathbb{P}}^{S_2}_V; D_{\mathcal{G}}^p) \simeq W_{p,\lambda}(\tilde{\mathbb{P}}^{S_1}_V, \tilde{\mathbb{P}}^{S_2}_V)^p. \quad (6)$$

| | Validation Set | Test Set | | | | |
|---|---|---|---|---|---|---|
| Model | STS16 | STS12 | STS13 | STS14 | STS15 | Avg. |
| BoW | 22.6 | 23.8 | 20.2 | 29.4 | 31.5 | 26.2 |
| Skip-thought | NA | 30.8 | 24.8 | 31.4 | 31.0 | 29.5 |
| SIF ($a = 0.001$, no PC removed) | 22.7 | 32.9 | 21.4 | 33.4 | 37.8 | 31.4 |
| SIF ($a = 0.001$, PC removed) | 41.2 | 34.4 | 43.0 | 45.2 | 48.1 | 42.7 |
| SIF ($a = 0.0001$, PC removed) | **55.4** | 40.5 | **49.8** | 51.0 | 52.0 | 48.3 |
| CoMB ($\alpha$=0.15, $\beta$=0.5, $s$=1) | 47.4 | 44.9 | 48.1 | 50.1 | 52.9 | 49.0 |
| CoMB ($\alpha$=0.55, $\beta$=0.5, $s$=5) | 47.6 | **49.1** | 40.6 | 53.4 | 52.7 | 48.9 |
| CoMB ($\alpha$=0.55, $\beta$=1, $s$=5) | 49.1 | 48.3 | 41.5 | **53.6** | **53.3** | **49.2** |

Table 1: Performance of *Context Mover's Barycenters* (CoMB) and related baselines on the STS tasks using Toronto Book Corpus[3]. The numbers are average *Pearson correlation x 100* (with respect to groundtruth scores). CoMB outperforms the best SIF baseline on **3 out of 4** tasks in the test set and also leads to an overall improvement on average for several hyperparameter settings. It is also **1.5x** better than the SIF with no PC removed and Skip-thought, and twice as better than BoW. Here, $\alpha, \beta, s$ denote the PPMI smoothing, column normalization exponent (Eq. 10) and k-shift. Skip-thought scores are taken from Arora et al. (2017).

**Empirical Evaluation.** To evaluate CoMB as an effective sentence representation, we consider 24 datasets from SemEval semantic textual similarity (STS) tasks (Agirre et al., 2012; 2013; 2014; 2015; 2016). The objective here is to give a score of how similar two sentences are in their meanings.

As a ground metric ($D_{\mathcal{G}}$), we consider the Euclidean distance between the point estimates (embeddings) of words. We train the GloVe (Pennington et al., 2014) embeddings on the Toronto Book Corpus (Zhu et al., 2015), and in this process also obtain the histogram information needed for the distributional estimate. Since GloVe embeddings for similar words are constructed to be close in terms of cosine similarity for similar words, we find the representative points by performing K-means clustering with respect to this similarity for $K = 300$.

We benchmark[4] our performance against SIF (Smooth Inverse Frequency) from Arora et al. (2017) who regard it as a "simple but tough-to-beat baseline", Skip-thought (Kiros et al., 2015), as well as against the plain Bag of Words (BoW) averaging. Hyperparameters for both SIF and CoMB are tuned on STS16, and the best configurations so obtained are used for comparison on the other STS tasks.

Table 1 shows that, on all the tasks CoMB outperforms the best variant of SIF on 3 out of 4 tasks in the test set and leads to an overall gain. Please refer to Tables 6 and 7 in Appendix A for detailed results and discussion. Also, we encourage the reader to check out Section A.6 where we qualitatively analyse CoMB & SIF.

**Online computation.** The principal component removal in SIF, which plays a crucial role as evident from the Table 1, has an aspect that typically goes under the rug. In particular, these principal components are estimated based on embeddings of sentences in the test set. This way it cleverly utilizes the information shared across sentences in the test set of downstream tasks, but can act as a hindrance in the practical usecase. For instance, consider the probable scenario (like for a chatbot) where we have to resolve a query about the similarity of two sentences in an online manner. Here SIF would undergo a significant performance drop, but CoMB is devoid of any issues arising in the online mode while retaining its performance.

**Summary and further prospects.** Our focus in these experiments was to compare methods which can build up sentence representations by just obtaining the word vector information. Hence, we didn't consider unsupervised methods such as Sent2vec (Pagliardini et al., 2017), that are specifically trained to work well on sentence similarity. We observe that CoMB serves as a competitive sentence

---

[3]The SIF numbers reported in Arora et al. (2017) are on Common Crawl which has 840 billion tokens, and hence have some difference on Toronto Book Corpus containing just 0.98 billion tokens.

[4]We use SIF's publicly available implementation (`https://github.com/PrincetonML/SIF`) and use SentEval (Conneau & Kiela, 2018) for evaluating BoW and CoMB.

representation method and remains applicable in the online usecase. The empirical results are quite encouraging, given the fact that we haven't even utilized the important property of non-associativity for Wasserstein barycenters (i.e., $B_p(\mu, B_p(\nu, \xi)) \neq B_p(B_p(\mu, \nu), \xi)$). This implies that we can take into account the word order with various aggregation strategies, like parse trees, and build the sentence representation by recursively computing barycenters phrase by phrase, which although remains beyond the scope of this paper.

Overall, this highlights towards the advantage of having distributional estimates for words, that can be extended to give a meaningful representation of sentences via CoMB in a principled manner.

## 7 HYPERNYMY DETECTION

In linguistics, hypernymy is a relation between words (or sentences) where the semantics of one word (the *hyponym*) are contained within that of another word (the *hypernym*). A simple form of this relation is the *is-A* relation, e.g., *cat* is an *animal*. Hypernymy is a special case of the more general concept of lexical entailment, the detection of which is relevant for tasks such as Question Answering (QA). Given a database of lexical entailment relations containing, e.g., **is-A**(Roger Federer, tennis player) might help a QA system answer *"Who is Switzerland's most successful tennis player?"*.

The early unsupervised approaches for this task exploited different linguistic properties of hypernymy (Weeds & Weir, 2003; Kotlerman et al., 2010; Santus et al., 2014; Rimell, 2014). While most of these are count-based, point embedding based methods (Chang et al., 2017; Henderson & Popa, 2016) have become more popular in recent years. Other approaches represent words by Gaussian distributions with KL-divergence as a measure of entailment (Vilnis & McCallum, 2014; Athiwaratkun & Wilson, 2017). These methods have proven to be powerful, as they not only capture the semantics but also the uncertainty about the contexts in which the word appears.

Therefore, hypernymy detection is a great testbed to verify the effectiveness of our approach (and the particular formulation) to represent each entity by the distribution of its contexts. To be successful on this task, a method has to consider if all contexts of the hyponym can be encompassed within the contexts of the hypernym. It can't just get away by predicting words that are similar. Hence, it is natural to make use of the Context Mover's Distance (CMD), Eq. 3, but with an appropriate ground cost that measures entailment relations well.

For this purpose, we utilize a recently proposed method by (Henderson & Popa, 2016; Henderson, 2017) which explicitly models what information is known about a word, by interpreting each entry of the embedding as the degree to which a certain feature is present. Based on the logical definition of entailment they derive an operator measuring the entailment similarity between two so-called entailment vectors defined as follows: $\vec{v}_i \oslash \vec{v}_j = \sigma(-\vec{v}_i) \cdot \log \sigma(-\vec{v}_j)$, where the sigmoid $\sigma$ and $\log$ are applied component-wise on the embeddings $\vec{v}_i, \vec{v}_j$. Thus, we use as ground cost $D_{ij}^{\text{Hend.}} := -\vec{v}_i \oslash \vec{v}_j$. This asymmetric ground cost also shows that our framework can be flexibly used with an arbitrary cost function defined on the ground space.

For tuning the hyperparameters, we utilize the HypeNet training set of Shwartz et al. (2016) (from the random split),

**Evaluation.** In total, we evaluated our method on 10 standard datasets: BLESS (Baroni & Lenci, 2011), EVALution (Santus et al., 2015), Benotto (2015), Weeds et al. (2014), BIBLESS (Kiela et al., 2015), Baroni et al. (2012), Kotlerman et al. (2010), Levy et al. (2014), HypeNet-Test (Shwartz et al., 2016), and Turney & Mohammad (2015). As an evaluation metric, we use average precision AP@all Zhu (2004). Following Chang et al. (2017) we pushed any OOV (out-of-vocabulary) words in the test data to the bottom of the list, effectively assuming that the word pairs do not have a hypernym relation.

The foremost thing that we would like to check is the benefit of having a distributional estimate in comparison to just the point embeddings. Here, we observe that by employing CMD along with the entailment embeddings, leads to a significant boost on most of the datasets, except on Baroni and Turney, where the performance is still competitive with the other state of the art methods like Gaussian embeddings. The more interesting observation is that on some datasets (EVALution, HypeNet, LenciBenotto) we even outperform or match state-of-the-art performance (cf. Table 2), by simply using CMD together with this ground cost $D_{ij}^{\text{Hend.}}$ based on the entailment embeddings.

| | Validation Set | Test Set | | | | | | |
| --- | --- | --- | --- | --- | --- | --- | --- | --- |
| Method | HypeNet-Train | HypeNet-Test | EVALution | LenciBenotto | Weeds | Turney | Baroni | BIBLESS |
| GE + C | NA | 21.6 | 26.7 | 43.3 | 52.0 | 53.9 | 69.7 | NA |
| GE + KL | NA | 23.7 | 29.6 | 45.1 | 51.3 | 52.0 | 64.6 | NA |
| DIVE + C·$\Delta$S | NA | 32.0 | 33.0 | **50.4** | **65.5** | **57.2** | **83.5** | NA |
| $D^{\text{Hend.}}$ | 29.0 | 28.8 | 31.6 | 44.8 | 60.8 | 56.6 | 78.3 | 70.5 |
| CMD$_{K=200}$+$D^{\text{Hend.}}$ | 53.4 | 53.4 | **38.1** | 50.1 | 63.9 | 56.0 | 67.5 | 74.0 |
| CMD$_{K=250}$+$D^{\text{Hend.}}$ | **53.6** | **53.7** | 37.1 | 49.9 | 63.8 | 56.3 | 67.3 | **74.9** |

Table 2: Comparison of the entailment vectors from Henderson (2017) used alone ($D^{\text{Hend.}}$), and when used together with our Context Mover's Distance, (CMD$_K$, where $K$ is the number of clusters), in the form of ground cost $D^{\text{Hend.}}$. The two listed CMD variants are the ones with best validation performance, when $K$ is fixed to 200 and 250. For reference, this table also includes state-of-the-art methods, like Gaussian embeddings with cosine similarity (GE+C) or KL-divergence (GE+KL), and DIVE. Scores for GE+C, GE+KL, and DIVE + C·$\Delta S$ are taken from Chang et al. (2017) as we use the same evaluation setup[6]. The scores are **AP@all (%)**. More details about the training setup and results on other datasets (along with the effect of PPMI parameters) can be found in Section A.1, Section A.7.1 & Table A.8. Numbers in **bold** indicate the best score for that dataset and the ones underlined denote the second best.

Notably, this approach is not specific to the entailment vectors from Henderson (2017) and more accurate set of vectors might help additionally. Alternatively, this also suggests that using CMD along with a method that produces embedding vectors (specialized for measuring the degree of entailment) can be a potential way to further improve the performance of that method.

## 8 Conclusion

We advocate for representing entities by a distributional estimate on top of any given co-occurrence structure. For each entity, we jointly consider the histogram information (with its contexts) as well as the point embeddings of the contexts. We show how this enables the use of optimal transport over distributions of contexts. Our framework results in an efficient, interpretable and compositional metric to represent and compare entities (e.g. words) and groups thereof (e.g. sentences), while leveraging existing point embeddings. We demonstrate its performance on several NLP tasks such as sentence similarity and word entailment detection. Motivated by the empirical results on the selected tasks, applying the proposed framework on co-occurrence structures beyond NLP is a promising direction.

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

## A SUPPLEMENTARY MATERIAL

### A.1 EXPERIMENTAL DETAILS

**Sentence Representations.** While using the Toronto Book Corpus, we remove the errors caused by crawling and pre-process the corpus by filtering out sentences longer than 300 words, thereby removing a very small portion (500 sentences out of the 70 million sentences). We utilize the code[7] from GloVe for building the vocabulary of size 205513 (obtained by setting min_count=10) and the co-occurrence matrix (considering a symmetric window of size 10). Note that as in GloVe, the contribution from a context word is inversely weighted by the distance to the target word, while computing the co-occurrence. The vectors obtained via GloVe have 300 dimensions and were trained for 75 iterations at a learning rate of 0.005, other parameters being the default ones. The performance of these vectors from GloVe was verified on standard word similarity tasks.

**Hypernymy Detection.** The training of the entailment vector is performed on a Wikipedia dump from 2015 with 1.7B tokens that have been tokenized using the Stanford NLP library (Manning et al., 2014). In our experiments, we use a vocabulary with a size of 80'000 and word embeddings with 200 dimensions. We followed the same training procedure as described in Henderson (2017) and were able to reproduce their scores on the hypernymy detection task. For tuning the hyperparameters, we utilize the HypeNet training set of Shwartz et al. (2016) (from the random split), following the procedure indicated in Chang et al. (2017) for tuning DIVE and Gaussian embeddings.

### A.2 PPMI DETAILS

**Formulation and Variants.** Typically, the probabilities used in PMIare estimated from the co-occurrence counts $\#(w, c)$ in the corpus and lead to

$$\text{PPMI}(w, c) = \max\left(\log\left(\frac{\#(w, c) \times |Z|}{\#(w) \times \#(c)}\right), 0\right), \tag{7}$$

where, $\#(w) = \sum_c \#(w, c)$, $\#(c) = \sum_w \#(w, c)$ and $|Z| = \sum_w \sum_c \#(w, c)$. Also, it is known that PPMI is biased towards infrequent words and assigns them a higher value. A common solution is to smoothen[8] the context probabilities by raising them to an exponent of $\alpha$ lying between 0 and 1. Levy & Goldberg (2014b) have also suggested the use of the shifted PPMI (SPPMI) matrix where the shift[9] by $\log(s)$ acts like a prior on the probability of co-occurrence of target and context pairs. These variants of PPMI enable us to extract better semantic associations from the co-occurrence matrix. Finally, we have

$$\text{SPPMI}_{\alpha,s}(w, c) := \max\left(\log\left(\frac{\#(w, c) \times \sum_{c'} \#(c')^\alpha}{\#(w) \times \#(c)^\alpha}\right) - \log(s), 0\right).$$

**Computational aspect.** We utilize the sparse matrix support of Scipy[10] for efficiently carrying out all the PPMI computations.

**PPMI Column Normalizations.** In certain cases, when the PPMI contributions towards the partitions (or clusters) have a large variance, it can be helpful to consider the fraction of $\mathcal{C}_k$'s SPPMI (Eq. 8, 9) that has been used towards a word $w$, instead of aggregate values used in 4. Otherwise the process of making the histogram unit sum might misrepresent the actual underlying contribution. We call this PPMI column normalization ($\beta$). In other words, the intuition is that the normlization will balance the effect of a possible non-uniform spread in total PPMI across the clusters. We observe that setting $\beta$ to 0.5 or 1 help in boosting performance on the STS tasks. The basic form of column

---

[7] https://github.com/stanfordnlp/GloVe

[8] $p_\alpha(c) := \frac{\#(c)^\alpha}{\sum_{c'} \#(c')^\alpha}$.

[9] Here, we denote the shift parameter by $s$ instead of the $k$ defined in (Levy et al., 2015) to avoid confusion with the other usage of $k$.

[10] https://docs.scipy.org/doc/scipy/reference/sparse.html

normalization is shown in 9.

$$(\tilde{\mathbf{H}}^w)_k := \frac{(\bar{\mathbf{H}}^w)_k}{\sum_{k=1}^{K} (\bar{\mathbf{H}}^w)_k} \quad \text{with} \tag{8}$$

$$(\bar{\mathbf{H}}^w)_k := \frac{\text{SPPMI}_{\alpha,s}(w, \mathcal{C}_k)}{\sum_w \text{SPPMI}_{\alpha,s}(w, \mathcal{C}_k)}. \tag{9}$$

Another possibility while considering the normalization to have an associated parameter $\beta$ that can interpolate between the above normalization and normalization with respcect to cluster size.

$$(\tilde{\mathbf{H}}_{\beta}^w)_k := \frac{(\bar{\mathbf{H}}_{\beta}^w)_k}{\sum_{k=1}^{K} (\bar{\mathbf{H}}_{\beta}^w)_k}, \quad \text{where}$$

$$(\bar{\mathbf{H}}_{\beta}^w)_k := \frac{\text{SPPMI}_{\alpha,s}(w, \mathcal{C}_k)}{\sum_w \text{SPPMI}_{\alpha,s}(w, \mathcal{C}_k)^{\beta}} \tag{10}$$

In particular, when $\beta = 1$, we recover the equation for histograms as in 9, and $\beta = 0$ would imply normalization with respect to cluster sizes.

### A.3 OPTIMAL TRANSPORT

**Implementation aspects.** We make use of the Python Optimal Transport (POT)[11] for performing the computation of Wasserstein distances and barycenters on CPU. For more efficient GPU implementation, we built custom implementation using PyTorch. We also implement a batched version for barycenter computation, which to the best of our knowledge has not been done in the past. The batched barycenter computation relies on a viewing computations in the form of block-diagonal matrices. As an example, this batched mode can compute around 200 barycenters in 0.09 seconds, where each barycenter is of 50 histograms (of size 100) and usually gives a speedup of about 10x.

**Scalability.** For further scalability, an alternative is to consider *stochastic optimal transport* techniques (Genevay et al., 2016). Here, the idea would be to randomly sample a subset of contexts from the distributional estimate while considering this transport.

**Stability of Sinkhorn Iterations.** For all our computations involving optimal transport, we typically use $\lambda$ around 0.1 and make use of log or median normalization as common in POT to stabilize the Sinkhorn iterations. Also, we observe that clipping the ground metric matrix (if it exceeds a particular large threshold) also sometimes results in performance gains.

**Value of $p$.** It has been shown in Agueh & Carlier (2011) that when the underlying space is Euclidean and $p = 2$, there exists a unique minimizer to the Wasserstein barycenter problem. But, since we are anyways solving the regularized Wasserstein barycenter (Cuturi & Doucet, 2014) problem over here instead of the exact one, the particular value of $p$ seems less of an issue. Empirically in the sentence similarity experiments, we have observed $p = 1$ to perform better than $p = 2$ (by about 2-3 points).

### A.4 CLUSTERING.

For clustering, we make use of kmcuda's[12] efficient implementation of K-Means algorithm on GPUs.

### A.5 SOFTWARE RELEASE

**Core code and histograms.** We plan to make all our code (for all these parts) and our pre-computed histograms (for the mentioned datasets) publicly available on GitHub soon.

---

[11] http://pot.readthedocs.io/en/stable/
[12] https://github.com/src-d/kmcuda

**Standard evaluation suite for Hypernymy.** To ease the evaluation pipeline, we have collected the most common benchmark datasets and compiled the code for assessing a model's performance on hypernymy detection or directionality into a Python package, called **HypEval**, which is publicly available. This also handles OOV (out-of-vocabulary) pairs in a standardized manner and allows for efficient, batched evaluation on GPU.

## A.6 Qualitative Analysis of Sentence Representations

Here, we would like to qualitatively probe the kind of results obtained when computing Wasserstein barycenter of the distributional estimates, in particular, when using CoMB to represent sentences. To this end, we consider a few simple sentences and find the closest word in the vocabulary for CoMB (with respect to CMD) and contrast it to SIF with cosine distance.

| Query | **CoMB** (with CMD) | **SIF** (with cosine, no PC removal) |
|---|---|---|
| ['i', 'love', 'her'] | love, hope, *always*, *actually*, *because*, *doubt*, *imagine*, *but*, *never*, *simply* | love, loved, breep-breep, *want*, clash-clash-clang, thysel, *know*, think, nope, *life* |
| ['my', 'favorite', 'sport'] | sport, *costume*, circus, *costumes*, *outfits*, super, sports, *tennis*, *brand*, fabulous | favorite, favourite, sport, wiccan-type, *pastime*, pastimes, sports, best, *hangout*, spectator |
| ['best', 'day', 'of', 'my', 'life'] | best, *for*, *also*, only, or, *anymore*, *all*, *is*, *having*, *especially* | life, day, best, c.5, writer/mummy, days, margin-bottom, time, margin-left,night |
| ['he', 'lives, 'in', 'europe', 'for'] | america, europe, *decades*, asia, *millenium*, preserve, *masters*, *majority*, elsewhere, *commerce* | lives, europe, life, america, lived, world, england, france, people, c.5 |
| ['he', 'may', 'not', 'live'] | *unless*, *perhaps*, must, may, *anymore*, will, likely, youll, would, certainly | may, live, should, will, might, must, margin-left, henreeeee, 0618082132, think |
| ['can', 'you', 'help', me', 'shopping'] | *anytime*, *yesterday*, *skip*, *overnight*, *wed*, *afterward*, choosing, figuring, deciding, shopping | help, can, going, want, *go*, *do*, think, need, able, take |
| ['he', 'likes', 'to', 'sleep', 'a', 'lot'] | *whenever*, forgetting, *afterward*, *pretending*, rowan, eden, *casper*, nash, annabelle, savannah, | lot, sleep, much, *besides*, better, likes, *really*, think, *probably*, talk |

Table 3: Top 10 closest neighbors for CoMB and SIF (no PC removed) found across the vocabulary, and sorted in ascending order of distance from the query sentence. Words in *italics* are those which in our opinion would fit well when added to one of the places in the query sentence. Note that, both CoMB (under current formulation) and SIF don't take the word order into account.

**Observations.** We find that closest neighbors (see Table 3 ) for CoMB consist of relatively more diverse set of words which fit well in the context of given sentence. For example, take the sentence "i love her", where CoMB captures a wide range of contexts, for example, "i *actually* love her", "i love her *because*", "i *doubt* her love" and more. Also for an ambiguous sentence "he lives in europe for", the obtained closest neighbors for CoMB include: 'decades', 'masters', 'majority', 'commerce' , etc., while with SIF the closest neighbors are mostly words similar to one of the query words. Further, if you look at the last three sentences in the Table 3, the first closest neighbor for CoMB even acts as a good next word for the given query. This suggests that CoMB might perform well on the task of sentence completion, but this additional evaluation is beyond the scope of this paper.

## A.7 DETAILED RESULTS

Detailed results of the hypernymy detection and sentence representation experiments are listed on the following pages.

### A.7.1 HYPERNYMY DETECTION

| Method | Dataset | | | | | |
|---|---|---|---|---|---|---|
| | BLESS | EVALution | LenciBenotto | Weeds | BIBLESS | Baroni |
| Henderson et al. ($D^{\text{Hend.}}$) | **6.4** | 31.6 | 44.8 | 60.8 | 70.5 | **78.3** |
| CMD ($K$=200) + $D^{\text{Hend.}}$ | 5.8 | **38.1** | **50.1** | **63.9** | 74.0 | **67.5** |
| CMD ($K$=250) + $D^{\text{Hend.}}$ | 5.8 | 37.1 | 49.9 | 63.8 | **74.9** | 67.3 |

| Method | Dataset | | | | |
|---|---|---|---|---|---|
| | Kotlerman | Levy | HypeNet-Test | Turney | **Avg.Gain** |
| Henderson et al. ($D^{\text{Hend.}}$) | 34.0 | 11.7 | 28.8 | **56.6** | - |
| CMD ($K$=200) + $D^{\text{Hend.}}$ | **34.7** | 12.2 | 53.4 | 56.0 | +3.2 |
| CMD ($K$=250) + $D^{\text{Hend.}}$ | 34.4 | **12.9** | **53.7** | 56.3 | **+ 3.3** |

Table 4: Comparison of the entailment vectors alone (Hend.) and when used together with our Context Mover's Distance, CMD($K$) (where $K$ is the number of clusters), in the form of ground cost $D^{\text{Hend.}}$. We also indicate the average gain in performance across these 10 datasets by using CMD along with the entailment vectors. All scores are AP at all (%).

The above listed variants of CMD are the ones with best validation performance on HypeNet-Train (Shwartz et al., 2016). The other hyperparameters (common) for both of them are as follows:

- PPMI smoothing, $\alpha = 0.5$.
- PPMI column normalization exponent, $\beta$=0.5.
- PPMI k-shift, $s$=1.
- Regularization constant for Wasserstein distance, $\lambda$=0.1
- Number of Sinkhorn iterations = 500.
- Log normalization of Ground Metric.

**Out of Vocabulary Details.**

| Dataset | Number of pairs (N) | Out of vocabulary pairs (OOV) |
|---|---|---|
| BLESS | 26554 | 1504 |
| EVALution | 13675 | 92 |
| LenciBenotto | 5010 | 1172 |
| Weeds | 2928 | 354 |
| BIBLESS | 1668 | 33 |
| Baroni | 2770 | 37 |
| Kotlerman | 2940 | 172 |
| Levy | 12602 | 4926 |
| HypeNet-Test | 17670 | 11334 |
| Turney | 2188 | 173 |

Table 5: Dataset sizes. N is the number of word pairs in the dataset, and OOV denotes how many word pairs are not processed.

### A.7.2 SENTENCE REPRESENTATION

We observe empirically that the PPMI smoothing parameter $\alpha$, which balances the bias of PPMI towards rare words, plays an important role. While its ideal value would vary on each task, we found the settings mentioned in the Table 6 to work well uniformly across the above spectrum of tasks.

| | STS12 | | | | |
|---|---|---|---|---|---|
| Model | MSRpar | MSRvid | SMTeuroparl | WordNet | SMTnews |
| BoW | 19.3 | 0.2 | 26.6 | 37.1 | 35.6 |
| SIF ($a = 0.001$, no PC removed) | 19.5 | 41.7 | 24.3 | 54.0 | 25.0 |
| SIF ($a = 0.001$, PC removed) | 21.0 | 36.5 | 31.0 | 55.4 | 27.9 |
| SIF ($a = 0.0001$, PC removed) | 20.1 | 58.8 | 31.2 | **55.8** | 36.9 |
| CoMB ($\alpha$=0.15, $\beta$=0.5, $s$=1) | 31.6 | **68.2** | 39.0 | 51.4 | 34.4 |
| CoMB ($\alpha$=0.55, $\beta$=0.5, $s$=5) | **32.6** | 63.6 | **48.8** | 53.4 | **47.1** |
| CoMB ($\alpha$=0.55, $\beta$=1, $s$=5) | 31.3 | 62.1 | 47.8 | 53.7 | 46.5 |

| | STS13 | | |
|---|---|---|---|
| Model | FNWN | Headlines | WordNet |
| BoW | 18.4 | 25.8 | 16.2 |
| SIF ($a = 0.001$, no PC removed) | 11.5 | 46.1 | 6.8 |
| SIF ($a = 0.001$, PC removed) | 14.3 | 54.3 | 60.4 |
| SIF ($a = 0.0001$, PC removed) | 13.5 | **60.5** | **75.5** |
| CoMB ($\alpha$=0.15, $\beta$=0.5, $s$=1) | **20.6** | 53.7 | 69.9 |
| CoMB ($\alpha$=0.55, $\beta$=0.5, $s$=5) | 6.3 | 53.5 | 62.1 |
| CoMB ($\alpha$=0.55, $\beta$=1, $s$=5) | 11.5 | 53.7 | 59.4 |

| | STS14 | | | | | |
|---|---|---|---|---|---|---|
| Model | Forum | News | Headlines | Images | WordNet | Twitter |
| BoW | 15.2 | 39.7 | 25.8 | 22.9 | 33.5 | 39.0 |
| SIF ($a = 0.001$, no PC removed) | 15.8 | 31.7 | 44.6 | 38.0 | 26.7 | 43.6 |
| SIF ($a = 0.001$, PC removed) | 15.2 | 35.7 | 52.1 | 47.4 | 62.6 | **58.0** |
| SIF ($a = 0.0001$, PC removed) | 23.3 | 43.0 | **57.0** | **52.8** | **76.4** | 53.8 |
| CoMB ($\alpha$=0.15, $\beta$=0.5, $s$=1) | 33.7 | 58.2 | 46.1 | 46.2 | 65.2 | 51.2 |
| CoMB ($\alpha$=0.55, $\beta$=0.5, $s$=5) | 32.1 | 62.7 | 48.7 | 51.0 | 67.2 | 55.9 |
| CoMB ($\alpha$=0.55, $\beta$=1, $s$=5) | **35.0** | **64.1** | 50.1 | 50.4 | 64.2 | 57.8 |

Table 6: Detailed **test set performance** of *Context Mover's Barycenters* (CoMB) and related baselines on the STS12, STS13, and STS14 tasks using Toronto Book Corpus. The numbers are average Pearson correlation x 100 (with respect to groundtruth scores). Here, $\alpha, \beta, s$ denote the PPMI smoothing, column normalization exponent (Eq. 10), and k-shift.

| | STS15 | | | | |
|---|---|---|---|---|---|
| Model | Forum | Students | Belief | Headlines | Images |
| BoW | 20.1 | 45.4 | 24.4 | 36.5 | 31.2 |
| SIF ($a = 0.001$, no PC removed) | 26.4 | 38.3 | 31.6 | 52.3 | 40.4 |
| SIF ($a = 0.001$, PC removed) | 30.0 | 62.0 | 39.0 | 59.1 | 50.6 |
| SIF ($a = 0.0001$, PC removed) | 34.0 | 63.7 | **48.4** | **62.4** | 51.7 |
| CoMB ($\alpha$=0.15, $\beta$=0.5, $s$=1) | **44.7** | 58.4 | 43.2 | 60.0 | 58.4 |
| CoMB ($\alpha$=0.55, $\beta$=0.5, $s$=5) | 39.0 | **63.3** | 37.8 | 60.3 | **63.1** |
| CoMB ($\alpha$=0.55, $\beta$=1, $s$=5) | 36.8 | 63.0 | 44.5 | 60.7 | 61.4 |

Table 7: (continued from Table 6) Detailed **test set performance** of *Context Mover's Barycenters* (CoMB) and related baselines on the STS15 using Toronto Book Corpus. The numbers are average Pearson correlation x 100 (with respect to groundtruth scores). Here, $\alpha, \beta, s$ denote the PPMI smoothing, column normalization exponent (Eq. 10), and k-shift.

| | STS16 | | | | |
|---|---|---|---|---|---|
| Model | Answer | Headlines | Plagiarism | Postediting | Question |
| BoW | 17.1 | 33.5 | 25.8 | 37.1 | -0.6 |
| SIF ($a = 0.001$, no PC removed) | 21.3 | 49.1 | 14.2 | 35.5 | -6.4 |
| SIF ($a = 0.001$, PC removed) | 26.0 | 57.0 | 43.4 | 61.5 | 18.2 |
| SIF ($a = 0.0001$, PC removed) | **34.2** | **60.2** | **58.0** | **71.2** | **53.5** |
| CoMB ($\alpha$=0.15, $\beta$=0.5, $s$=1) | 21.6 | 51.9 | 48.8 | 64.0 | 50.9 |
| CoMB ($\alpha$=0.55, $\beta$=0.5, $s$=5) | 18.0 | 53.0 | 54.6 | 65.6 | 46.7 |
| CoMB ($\alpha$=0.55, $\beta$=1, $s$=5) | 26.2 | 54.8 | 51.3 | 66.6 | 46.6 |

Table 8: Detailed **validation set performance** of *Context Mover's Barycenters* (CoMB) and related baselines on the STS16 using Toronto Book Corpus. The numbers are average Pearson correlation x 100 (with respect to groundtruth scores). Note that, STS16 was used as the validation set to obtain the best hyperparameters for all the methods in these experiments. As a result, high performance on STS16 may not be indicative of the overall performance. Here, $\alpha, \beta, s$ denote the PPMI smoothing, column normalization exponent (Eq. 10), and k-shift.

## A.8 EFFECT OF PPMI PARAMETERS FOR HYPERNYMY DETECTION

| | Dataset | | | | | |
|---|---|---|---|---|---|---|
| Method | BLESS | EVALution | LenciBenotto | Weeds | BIBLESS | Baroni |
| Henderson et al. ($D^{\text{Hend.}}$) | 6.4 | 31.6 | 44.8 | 60.8 | 70.5 | **78.3** |
| CMD ($\alpha$=0.15, $s$=1) + $D^{\text{Hend.}}$ | **7.3** | 37.7 | 49.0 | 63.6 | 74.8 | 64.4 |
| CMD ($\alpha$=0.15, $s$=5) + $D^{\text{Hend.}}$ | 6.9 | 39.1 | 49.4 | 64.3 | 74.0 | 65.2 |
| CMD ($\alpha$=0.15, $s$=15) + $D^{\text{Hend.}}$ | 7.0 | 39.8 | 48.5 | 64.7 | 75.0 | 65.6 |
| CMD ($\alpha$=0.5, $s$=1) + $D^{\text{Hend.}}$ | 6.6 | 39.2 | 48.6 | 62.9 | **76.1** | 64.6 |
| CMD ($\alpha$=0.5, $s$=5) + $D^{\text{Hend.}}$ | 5.9 | 40.4 | **49.9** | 65.7 | 73.9 | 67.2 |
| CMD ($\alpha$=0.5, $s$=15) + $D^{\text{Hend.}}$ | 5.5 | **40.5** | 49.5 | **66.2** | 72.8 | 67.4 |

| | Dataset | | | | |
|---|---|---|---|---|---|
| Method | Kotlerman | Levy | Turney | **Avg.Gain** | **Avg. Gain (w/o Baroni)** |
| Henderson et al. ($D^{\text{Hend.}}$) | 34.0 | 11.7 | 56.6 | - | - |
| CMD ($\alpha$=0.15, $s$=1) + $D^{\text{Hend.}}$ | 33.9 | 10.8 | 57.2 | +0.5 | +2.2 |
| CMD ($\alpha$=0.15, $s$=5) + $D^{\text{Hend.}}$ | 34.2 | 11.6 | 57.0 | +0.8 | +2.5 |
| CMD ($\alpha$=0.15, $s$=15) + $D^{\text{Hend.}}$ | 34.9 | 12.3 | **57.3** | +1.2 | **+2.9** |
| CMD ($\alpha$=0.5, $s$=1) + $D^{\text{Hend.}}$ | 34.7 | 10.2 | 56.8 | +0.6 | +2.4 |
| CMD ($\alpha$=0.5, $s$=5) + $D^{\text{Hend.}}$ | 34.6 | 11.3 | 56.5 | +1.2 | +2.7 |
| CMD ($\alpha$=0.5, $s$=15) + $D^{\text{Hend.}}$ | **35.6** | **12.6** | 56.1 | **+1.3** | +2.8 |

Table 9: Comparison of the entailment vectors alone (Hend.) and when used together with our Context Mover's Distance, CMD($\alpha$, $s$) (where $\alpha$ and $s$ are the PPMI smoothing and shift parameters), in the form of ground cost $D^{\text{Hend.}}$. All of the CMD variants use $K = 100$ clusters. We observe that using our method with the entailment vectors performs better on 8 out of 9 datasets in comparsion to just using these vectors alone. Avg. gain refers to the average gain in performance relative to the entailment vectors. Avg. gain w/o Baroni refers to the average performance gain excluding the Baroni dataset. The hyperparameter $\alpha$ refers to the smoothing exponent and $s$ to the shift in the PPMI computation. All scores are AP at all (%).

This table was generated during an earlier version of the paper, when we were not considering the validation on HypeNet-Train. Hence, the above table doesn't contain numbers on HypeNet-Test, but an indication of performance on it can be seen in Section A.7.1. In any case, this table suggests that our method works well for several PPMI hyper-parameter configurations.

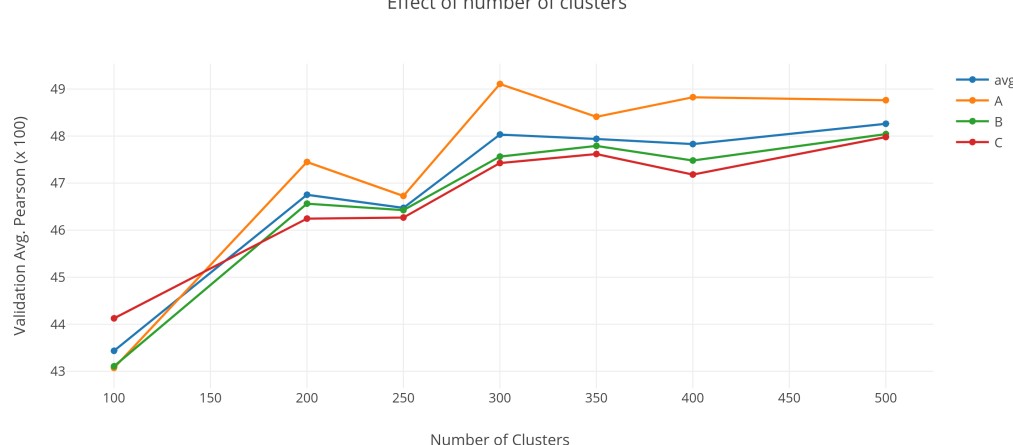

Figure 3: Effect of the number of clusters ($K$) on validation performance. A, B, C correspond to the three best performing variants of CoMB obtained as per validation on STS16 and as presented in the Table 1. In particular, A denotes the hyperparameter setting of [($\alpha$=0.55, $\beta$=1, $s$=5], B refers to [$\alpha$=0.55, $\beta$=0.5, $s$=5] and C denotes [$\alpha$=0.15, $\beta$=0.5, $s$=1]. The *'avg'* plot shows the average trend across these three configurations.

## A.9 EFFECT OF NUMBER OF CLUSTERS

Here, we analyse the impact of number of clusters on the performance of Context Mover's Barycenters (CoMB) for the sentence similarity experiments (c.f. Section 6). In particular, we look at the three best performing variants (A, B, C) on the validation set (STS 16) as well as averaged across them.

We observe in Figure 3 that on average the performance significantly improves when the number of clusters are increased until around $K = 300$, and beyond that mostly plateaus ($\pm$ 0.5). But, as can be seen for variants B and C the performance typically continues to rise until $K = 500$. It seems that the amount of PPMI column normalization ($\beta = 0.5$ vs $\beta = 1$) might be at play here.

Overall, going from $K = 300$ to $K = 500$ comes at the cost of increased computation time, and doesn't lead to a substantial gain in performance. This reflects the tradeoff between accuracy and efficiency. Hence, we stick to $K = 300$ for our results on sentence similarity tasks.

Such a trend seems to be inline with the ideal case where we wouldn't need to do any clustering and just take all possible contexts into account. Thus, it suggests that better ways (other than clustering) to deal with this problem might further boost the performance.

# B   QUALITATIVE ANALYSIS OF SENTENCE SIMILARITY

In this section, we aim to qualitatively analyse the particular examples where our method, Context Mover's Barycenters (CoMB), performs better or worse than the Smooth Inverse Frequency (SIF) approach from Arora et al. (2017).

## B.1   EVALUATION PROCEDURE

**Comparing by rank.**   It doesn't make much sense to compare the raw distance values between two sentences as given by Context Mover's Distance (CMD) for CoMB and cosine distance for SIF. This is because the spread of distance values across sentence pairs can be quite different. Note that the quantitative evaluation of these tasks is also carried out by Pearson/Spearman rank correlation of the predicted distances/similarities with the ground-truth scores.

Thus, in accordance with this reasoning, we compare the similarity score of a sentence pair relative to its rank based on ground-truth score (amongst the sentence pairs for that dataset). So, the better method should rank sentence pairs closer to the ranking obtained via ground-truth scores.

| Ground-Truth Score | Implied meaning |
|---|---|
| 5 | The two sentences are completely equivalent, as they mean the same thing. |
| 4 | The two sentences are mostly equivalent, but some unimportant details differ. |
| 3 | The two sentences are roughly equivalent, but some important information differs/missing. |
| 2 | The two sentences are not equivalent, but share some details. |
| 1 | The two sentences are not equivalent, but are on the same topic. |
| 0 | The two sentences are completely dissimilar. |

Table 10: STS ground scores and their implied meanings, as taken from Agirre et al. (2015)

**Ground-truth details.**   The ground-truth scores (can be fractional) and range from 0 to 5, and the meaning implied by the integral score values can be seen in the Table 10. In the case where different examples have the same ground-truth score, the ground-truth rank is then based on lexicographical ordering of sentences for our qualitative evaluation procedure. (This for instance means that sentence pairs ranging from 62 to 74 would correspond to the same ground-truth score of 4.6). The ranking is done in the descending order of sentence similarity, i.e., most similar to least similar.

**Example selection criteria.**   For all the examples, we compare the best variants of CoMB and SIF on those datasets. We particularly choose those examples where there is the maximum difference in ranks according to CoMB and SIF, as they would be more indicative of where a method succeeds or fails. Nevertheless, such a qualitative evaluation is subjective and is meant to give a better understanding of things happening under the hood.

## B.2   EXPERIMENTS AND OBSERVATIONS

We look at examples from three datasets, namely: Images from STS15, News from STS14 and WordNet from STS14 to get a better idea of an overall behavior. In terms of aggregate quantitative performance, on Images and News datasets, CoMB is better than SIF, while the opposite is true for WordNet. These examples across the three datasets may not probably be exhaustive and are up to subjective interpretation, but hopefully will lend some indication as to where and why each method works.

### B.2.1   TASK: STS14, DATASET: NEWS

We look in detail at the examples in News dataset from STS 2014 (Agirre et al., 2014). The results of qualitative analysis on Images and WordNet datasets can be found in Section B.5. For reference, CoMB does a better job overall with a Pearson correlation (x100) of 64.1 versus 43.0 for SIF, as presented in Table 6. The main observations are:

**Observation 1.**   Examples 1, 2, 4, 5 are sentence pairs which are equivalent in meaning (c.f. Table 10), but typically have additional details in the predicates of the sentences. Here, CoMB is

| | Sentence 1 | Sentence 2 | Ground-Truth Score | Ground-Truth Ranking | CoMB Ranking | SIF Ranking |
|---|---|---|---|---|---|---|
| 1 | the united states government and other nato members have refused to ratify the amended treaty until officials in moscow withdraw troops from the former soviet republics of moldova and georgia . | the united states and other nato members have refused ratify the amended treaty until russia completely withdraws from moldova and georgia . | 4.6 | 30 | **67** | 152 |
| 2 | jewish-american group the anti-defamation league ( adl ) published full-page advertisements in swiss and international papers in april 2008 accusing switzerland of funding terrorism through the deal . | the anti-defamation league took out full-page advertisments in swiss and international newspapers earlier in april 2008 accusing switzerland of funding terrorism through the deal . | 4.4 | 36 | **35** | 128 |
| 3 | the judicial order accused raghad of funding terrorism . | the court accused raghad saddam hussein of funding terrorism . | 4.2 | 59 | 258 | **124** |
| 4 | estonian officials stated that some of the cyber attacks that caused estonian government websites to shut down temporarily came from computers in the administration of russia including in the office of president vladimir putin . | officials in estonia including prime minister andrus ansip have claimed that some of the cyber attacks came from russian government computers including computers in the office of russian president vladimir putin . | 3.8 | 86 | **84** | 206 |
| 5 | the african union has proposed a peace-keeping mission to help somalia ' s struggling transitional government stabilize somalia . | the african union has proposed a peace-keeping mission to aid the struggling transitional government in stabilizing somalia , particularly after the withdrawal of ethiopian forces | 3.6 | 119 | **104** | 262 |
| 6 | some asean officials stated such standardization would be difficult due to different countries ' political systems . | some officials stated the task would be difficult for asean members because of varied legal and political systems . | 3.6 | 117 | 244 | **108** |
| 7 | nicaragua commemorated the 25th anniversary of the sandinista revolution . | nicaragua has not reconciled how to approach the anniversary of the sandinista revolution . | 2.4 | 213 | **250** | 48 |
| 8 | south korea launches new bullet train reaching 300 kph . | south korea has had a bullet train system since the 1980s . | 2 | 232 | **267** | 130 |
| 9 | south korea and israel oppose proliferation of weapons of mass destruction and an arms race . | china will resolutely oppose the proliferation of mass destructive weapons . | 1.4 | 262 | 164 | **235** |
| 10 | china is north korea ' s closest ally . | north korea is a reclusive state . | 1.2 | 265 | **279** | 196 |
| 11 | the chinese government gave active cooperation and assistance to the organization for the prohibition of chemical weapons inspections . | the ecuadorian foreign ministry said in a statement that delegates from the organization for the prohibition of chemical weapons ( opaq ) will also take part in the meeting . | 1 | 277 | 158 | **231** |
| 12 | do quy doan is a spokesman for the vietnamese ministry of culture and information . | grenell is spokesman for the u.s. mission to the united nations . | 0.8 | 282 | 213 | **292** |

Table 11: Examples of some indicative sentence pairs, from *News* dataset in *STS14*, with ground-truth scores and ranking as obtained via (best variants of) CoMB and SIF. The total number of sentences is *300* and the ranking is done in descending order of similarity. The method which ranks an example closer to the ground-truth rank is better and is highlighted in **blue**. CoMB ranking is the one produced when representing sentences via CoMB and then using CMD to compare them. SIF ranking is when sentences are represented via SIF and then employing cosine similarity.

better than SIF at ranking the pairs closer to the ground-truth ranking. This probably suggests the averaging of word embeddings, which is the 1[st] step in SIF, is not as resilient to the presence of such details than the Wasserstein barycenter of distributional estimates in CoMB. We speculate that when having distributional estimates (where multiple senses or contexts are considered), adding details can help towards refining the particular meaning implied.

**Observation 2.** Let's consider the examples 3 and 6 where SIF is better than CoMB. These are sentence pairs which are equivalent or roughly equivalent in meanings, but with a few words substituted (typically subjects) like *"judicial order"* instead of *"court"* in example 3. Here it seems that the substitution is adverse for CoMB while considering varied senses through the distributional estimate, in comparison to looking at the "point" meaning given by SIF.

**Observation 3.** In 7, 8, and 10, each sentence pair is about a common topic, but the meaning of individual sentences is quite different. For instance, example 8: *"south korea launches new bullet train reaching 300 kph"* & *"south korea has had a bullet train system since the 1980s"*. Or like in example 10: *"china is north korea ' s closest ally"* & *"north korea is a reclusive state"*. Note that typically in these examples, the subject is same in a sentence pair, and the difference is mainly in the predicate. Here, CoMB identifies the difference and ranks them closer to the ground-truth. Whereas, SIF fails to understand this and ranks them as more similar (and far away) than the ground-truth.

**Observation 4.** The examples 9, 11, and 12 are related sentences and differ mainly in details such as the name of the country, person, department, i.e. proper nouns. In particular, consider example 9: *"south korea and israel oppose proliferation of weapons of mass destruction and an arms race"* & *"china will resolutely oppose the proliferation of mass destructive weapons"*. The main difference in these examples stems from differences in the subject rather than the predicate. CoMB considers these sentence pairs to be more similar than suggested by ground-truth. Hence, in such scenarios where the subject (like the particular proper nouns) makes the most difference, SIF seems to be better.

### B.3 CONCLUSIONS FROM QUALITATIVE EXAMPLES

Summarizing the observations from the above qualitative analysis on News dataset [13], we conclude the following about the nature of success or failures of each method.

- When the subject of the sentence is similar and main difference stems from the predicate, CoMB is the winner. This can be seen for both the case when predicates are equivalent but described distinctly *(observation 1)* and when predicates are not equivalent *(observation 3)*.

- When the predicates are similar and the distinguishing factor is in the subject (or object), SIF takes the lead. This seems to be true for both scenarios when the subject used increases or decreases the similarity as measured by CoMB, *(observations 2 and 4)*.

- The above two points in a way also signify where having distributional estimates can be better or worse than point estimates.

- CoMB and SIF appear to be complementary in the kind of errors they make. Hence, combining the two is an exciting future avenue.

Lastly, it also seems worthwhile to explore having different ground metrics for CoMB and CMD (which are currently shared). The ground metric plays a crucial role in performance and the nature of these observations. Employing a ground metric(s) that better handles the above subtleties would be a useful research direction.

---

[13]Similar findings can also be seen for the two other datasets in Section B.5.

B.4   EFFECT OF SENTENCE LENGTH

In this section, we look at the length of sentences across all the datasets in each of the STS tasks. Average sentence length is one measure of the complexity of a particular dataset. But looking at just sentence lengths may not give a complete picture, especially for the textual similarity tasks where there can be many words common between the sentence pairs. The Table 12 shows the various statistics of each dataset, with respect to the sentence lengths along with the better method on each of them (out of CoMB and SIF).

| Task-Dataset | # sentence pairs | Avg. sentence length | Avg. word overlap (per sentence pair) | Avg. effective sentence length (excluding common words) | Better method |
|---|---|---|---|---|---|
| STS12-MSRpar | 750 | 21.16 | 14.17 | 6.99 | CoMB |
| STS12-MSRvid | 750 | 7.65 | 4.70 | 2.95 | CoMB |
| STS12-SMTeuroparl | 459 | 12.33 | 8.11 | 4.22 | CoMB |
| STS12-WordNet | 750 | 8.82 | 5.03 | 3.79 | SIF |
| STS12-SMTnews | 399 | 13.62 | 8.66 | 4.96 | SIF |
| STS13-FNWN | 189 | 22.94 | 2.53 | 20.41 | CoMB |
| STS13-Headlines | 750 | 7.80 | 3.76 | 4.05 | SIF |
| STS13-WordNet | 561 | 8.17 | 4.64 | 3.53 | SIF |
| STS14-Forum | 450 | 10.48 | 7.03 | 3.45 | CoMB |
| STS14-News | 300 | 17.42 | 11.59 | 5.83 | CoMB |
| STS14-Headlines | 750 | 7.91 | 3.89 | 4.01 | SIF |
| STS14-Images | 750 | 10.18 | 6.20 | 3.98 | SIF |
| STS14-WordNet | 750 | 8.87 | 4.83 | 4.05 | SIF |
| STS14-Twitter | 750 | 12.25 | 4.85 | 7.40 | (equal) |
| STS15-Forum | 375 | 17.77 | 4.29 | 13.49 | CoMB |
| STS15-Students | 750 | 10.70 | 5.33 | 5.37 | CoMB |
| STS15-Belief | 375 | 16.53 | 6.27 | 10.26 | SIF |
| STS15-Headlines | 750 | 8.00 | 3.71 | 4.29 | SIF |
| STS15-Images | 750 | 10.66 | 6.07 | 4.59 | CoMB |

Table 12: Analysis of sentence lengths in each of the datasets from STS12, STS13, STS14, and STS15. Along with the average sentence lengths, we also measure average word overlap in the sentence pair and thus the average *effective sentence length (i.e., after excluding the overlapping/common words in the sentence pair)*. For reference, we also show which out of CoMB or SIF performs better. On STS14-Twitter, the difference in performance isn't significant and we thus write 'equal' in the corresponding cell.

**Observations.**

- We notice that on datasets with longer effective sentence lengths, CoMB performs better than SIF on average. There might be other factors at play here, but if one had to pick on the axis of effective sentence length, CoMB leads over SIF [14].

- The above statement also aligns well with the *observation 1* from the qualitative analysis (c.f. Section B.2.1), that having more details can help in refining the particular meaning or sense implied by CoMB. (Effective sentence length can serve as a good proxy for indicating the amount of details.)

- It also seems to explain why both methods don't perform well (see Table 6) on STS13-FNWN, which has on average the maximum effective sentence length (of 20.4).

- To an extent, it also points towards the effect of corpora. For instance, in a corpus such as WordNet, which has a low average sentence length and with examples typically concerned about word definitions (see Table 14), SIF seems to be better of the methods. On the other hand, CoMB seems to be better for News (Table 11), Image captions (Table 13) or Forum.

---

[14]Effective sentence length averaged across datasets where CoMB is better is **7.48**. Contrast this to an average effective sentence length of **5.03** across datasets where SIF is better.

## B.5 Additional Qualitative Analysis

### B.5.1 Task: STS15, Dataset: Images

We consider the sentence pairs from Images dataset in STS15 task (Agirre et al., 2015), as presented in Table 13. As a reminder, CoMB outperforms SIF on this dataset with a Pearson correlation (x100) of 63.1 versus 51.7, as mentioned in Table 7.. The main observations are:

| | Sentence 1 | Sentence 2 | Ground-Truth Score | Ground-Truth Ranking | CoMB Ranking | SIF Ranking |
|---|---|---|---|---|---|---|
| 1 | the man and two young boys jump on a trampoline . | a man and two boys are bouncing on a trampoline . | 4.8 | 68 | **74** | 640 |
| 2 | a boy waves around a sparkler . | a young boy is twisting a sparkler around in the air . | 4.4 | 126 | **195** | 624 |
| 3 | a dog jumps in midair to catch a frisbee . | the brown dog jumps for a pink frisbee . | 4 | 184 | **161** | 481 |
| 4 | a child is walking from one picnic table to another . | the boy hops from one picnic table to the other in the park . | 3.2 | 287 | **401** | 737 |
| 5 | three boys are running on the beach playing a game . | two young boys and one young man run on a beach with water behind them . | 3.2 | 306 | **260** | 421 |
| 6 | a boy swinging on a swing . | the girl is on a swing . | 2.4 | 380 | **410** | 622 |
| 7 | a man is swinging on a rope above the water . | a man in warm clothes swinging on monkey bars at night . | 1.6 | 492 | 259 | **606** |
| 8 | a skier wearing blue snow pants is flying through the air near a jump . | a skier stands on his hands in the snow in front of a movie camera . | 1.4 | 514 | 264 | **605** |
| 9 | two black and white dogs are playing together outside . | two children and a black dog are playing out in the snow . | 1 | 570 | 185 | **372** |
| 10 | three dogs running in the dirt . | the yellow dog is running on the dirt road . | 1 | 524 | 303 | **531** |
| 11 | a little girl and a little boy hold hands on a shiny slide . | a little girl in a paisley dress runs across a sandy playground . | 0.4 | 629 | **683** | 354 |
| 12 | a little girl walks on a boardwalk with blue domes in the background . | a man going over a jump on his bike with a river in the background . | 0 | 696 | 310 | **591** |

Table 13: Examples of some indicative sentence pairs, from *Images* dataset in *STS15*, with ground-truth scores and ranking as obtained via (best variants of) CoMB and SIF. The total number of sentences is **750** and the ranking is done in descending order of similarity. The method which ranks an example closer to the ground-truth rank is better and is highlighted in **blue**. CoMB ranking is the one produced when representing sentences via CoMB and then using CMD to compare them. SIF ranking is when sentences are represented via SIF and then employing cosine similarity.

**Observation A.** Example 1 to 5 indicate pairs of sentences which are essentially equivalent in meaning, but with varying degrees of equivalence. Here, we can see that CoMB with CMD is able to rank the similarity between these pairs quite well in comparison to SIF, even when their way of describing is different. For instance, example 2 : *"a boy waves around a sparkler"* & *"a young boy is twisting a sparkler around in the air"*. This points towards the benefit of having multiple senses or contexts encoded through the distributional estimate in CoMB.

**Observation B.** Next, in the examples 7 to 10, which consist of sentence pairs that are not equivalent but have commonalities (about the topic). Here, SIF ranks the sentences closer to the ground-truth ranking while CoMB interprets these pairs as being more common in meaning than given by ground-truth. This could be the consequence of comparing the various senses or contexts implied by the sentence pairs via CMD. Take for instance, example 10, *"three dogs running in the dirt"* & *"the yellow dog is running on the dirt road"*. Since these sentences are about the similar topic (and the major difference is in their subject), this can result in CMD considering them more similar than cosine distance.

**Observation C.** For sentences which are completely dissimilar as per ground-truth, let's look at example 11 and 12. Consider 11, which is *"a little girl and a little boy hold hands on a shiny slide"*

& *"a little girl in a paisley dress runs across a sandy playground",* the sentences meaning totally different things and CoMB seems to be better at ranking than SIF. But, consider example 12: *"a little girl walks on a boardwalk with blue domes in the background"* & *"a man going over a jump on his bike with a river in the background"*. One common theme[15] can be thought as *"a person moving with something blue in the background"*, which can result in CoMB ranking the sentence as more similar. SIF also ranks it higher (at 591) than ground-truth (696), but is more closer than CoMB which ranks it at 310.

### B.5.2   TASK: STS14, DATASET: WORDNET

Lastly, we discuss the examples and observations derived from the qualitative analysis on WordNet dataset from STS14 (Agirre et al., 2014). This dataset is comprised of sentences which are the definitions of words/phrases, and sentence length is typically smaller than the datasets discussed before. For reference, SIF (76.4) does better than CoMB (67.2) in terms of average Pearson correlation (x100), as mentioned in Table 6.

|  | Sentence 1 | Sentence 2 | Ground-Truth Score | Ground-Truth Ranking | CoMB Ranking | SIF Ranking |
|---|---|---|---|---|---|---|
| 1 | combine so as to form a more complex product . | combine so as to form a whole ; mix . | 4.6 | 127 | **142** | 335 |
| 2 | ( cause to ) sully the good name and reputation of . | charge falsely or with malicious intent ; attack the good name and reputation of someone . | 4.4 | 176 | **235** | 534 |
| 3 | a person or thing in the role of being a replacement for something else | a person or thing that takes or can take the place of another . | 4.2 | 248 | **270** | 535 |
| 4 | create something in the mind . | form a mental image of something that is not present or that is not the case . | 3.6 | 340 | **443** | 683 |
| 5 | the act of surrendering an asset | the act of losing or surrendering something as a penalty for a mistake or fault or failure to perform etc . | 3 | 405 | **445** | 639 |
| 6 | ( attempt to ) convince to enroll , join or participate | register formally as a participant or member . | 2.8 | 406 | **423** | 507 |
| 7 | return to a prior state . | return to an original state . | 4.4 | 219 | 384 | **231** |
| 8 | give away something that is not needed . | give up what is not strictly needed . | 4.2 | 261 | 709 | **383** |
| 9 | a person who is a member of the senate . | a person who is a member of a partnership . | 0.4 | 553 | 260 | **429** |
| 10 | the context or setting in which something takes place . | the act of starting something . | 0 | 717 | 485 | **707** |
| 11 | a spatial terminus or farthest boundary of something . | a relation that provides the foundation for something . | 0 | 620 | 500 | **623** |
| 12 | the act of beginning something new . | the act of rejecting something . | 0 | 670 | **677** | 539 |

Table 14: Examples of some indicative sentence pairs, from *WordNet* dataset in *STS14*, with ground-truth scores and ranking as obtained via (best variants of) CoMB and SIF. The total number of sentences is **750** and the ranking is done in descending order of similarity. The method which ranks an example closer to the ground-truth rank is better and is highlighted in **blue**. CoMB ranking is the one produced when representing sentences via CoMB and then using CMD to compare them. SIF ranking is when sentences are represented via SIF and then employing cosine similarity.

**Observation D.**   Consider examples 1 to 6 as shown in Table 14, which fall in the category of equivalent sentences but in varying degrees. The sentence pairs essentially indicate different ways of characterizing equivalent things. Here, CoMB is able to rank the similarity between sentences in a better manner than SIF. Specifically, see example 2: *"( cause to ) sully the good name and reputation of"* & *"charge falsely or with malicious intent ; attack the good name and reputation of someone"*. It seems that SIF is not able to properly handle the additional definition present in sentence 2 and ranks this pair much lower in similarity at 534 versus 235 for CoMB. This is also in line with observation 1 about added details in the Section B.2.1.

---

[15]Of course, this is upto subjective interpretation.

**Observation E.**  In the examples 7 to 9, where CoMB doesn't do well in comparison to SIF, mainly have a slight difference in the object of the sentence. For instance, in example 9: *"a person who is a member of the senate"* & *"a person who is a member of a partnership"*. So based on the kind of substituted word, looking at its various contexts via the distributional estimate can make it more or less similar than desired. In such cases, using the "point" meanings of the objects seems to fare better. This also aligns with the observations 2 and 4 in the Section B.2.1.

## C    QUALITATIVE ANALYSIS OF HYPERNYMY DETECTION

Here, our objective is to qualitatively analyse the particular examples where our method of using Context Mover's Distance (CMD) along with embeddings from Henderson (2017) performs better or worse than just using these entailment embeddings alone.

### C.1    EVALUATION PROCEDURE

**Comparing by rank.**  Again as in the qualitative analysis with sentence similarity, it doesn't make much sense to compare the raw distance/similarity values between two words as their spread across word pairs can be quite different. We thus compare the ranks assigned to each word pair by both the methods.

**Ground-truth details.**  In contrast to graded ground-truth scores in the previous analysis, here we just have a binary ground truth: 'True' if the hyponym-hypernym relation exists and 'False' when it doesn't. We consider the BIBLESS dataset (Kiela et al., 2015) for this analysis, which has a total of 1668 examples. Out of these, 33 word pairs are not in the vocabulary (see Table 5), so we ignore them for this analysis. Amongst the 1635 examples left, 814 are 'True' and 821 are 'False'. A perfect method should rank the examples labeled as 'True' from 1 to 814 and the 'False' examples from 815 to 1635. Of course, achieving this is quite hard, but the better of the methods should rank as many examples in the desired ranges.

**Example selection criteria.**  We look at the examples where the difference in ranks as per the two methods is the largest. Also, for a few words, we also look at how each method ranks when present as a hypernym and a hyponym. If the difference in ranks is defined as, *CMD rank - Henderson Rank*, we present the top pairs where this difference is most positive and most negative.

### C.2    RESULTS

For reference on the BIBLESS dataset, CMD performs better than Henderson embeddings quantitatively (c.f. Table 2). Let's take a look at some word pairs to get a better understanding.

#### C.2.1    MAXIMUM POSITIVE DIFFERENCE IN RANKS

These are essentially examples where CMD considers the entailment relation as 'False' while the Henderson embeddings predict it as 'True', and both are most certain about their decisions. Table 15 shows these pairs, along with ranks assigned by the two methods and the ground-truth label for reference.

Some quick observations: many of the word pairs which the Henderson method gets wrong are co-hyponym pairs, such as: ('banjo', 'flute'), ('guitar', 'trumpet'), ('turnip, 'radish'). Additionally, ('bass', 'cello' ), ('creature', 'gorilla'), etc., are examples where the method has to assess not just if the relation exists, but also take into account the directionality between the pair, which the Henderson method seems unable to do.

#### C.2.2    MAXIMUM NEGATIVE DIFFERENCE IN RANKS

Now the other way around, these are examples where CMD considers the entailment relation as 'True' while the Henderson embeddings predict it as 'False', and both are most certain about their decisions. Table 16 shows these pairs. The examples where CMD performs poorly like, ('box', 'mortality'), ('pistol', 'initiative') seem to be unrelated and we speculate that matching the various contexts or

| Hypernym candidate | Hypernym candidate | Ground Truth | CMD rank | Henderson rank | Better Method |
|---|---|---|---|---|---|
| bass | cello | FALSE | 1346 | 56 | CMD |
| banjo | flute | FALSE | 1312 | 108 | CMD |
| guitar | trumpet | FALSE | 1249 | 52 | CMD |
| trumpet | violin | FALSE | 1351 | 165 | CMD |
| gill | goldfish | FALSE | 1202 | 21 | CMD |
| topside | battleship | FALSE | 1508 | 345 | CMD |
| trumpet | piano | FALSE | 1289 | 126 | CMD |
| washer | dishwasher | FALSE | 1339 | 234 | CMD |
| gun | pistol | FALSE | 1270 | 166 | CMD |
| cauliflower | rainbow | FALSE | 1197 | 136 | CMD |
| hawk | woodpecker | FALSE | 1265 | 210 | CMD |
| garlic | spice | TRUE | 1248 | 204 | Henderson |
| coyote | beast | TRUE | 1096 | 57 | Henderson |
| lizard | beast | TRUE | 1231 | 201 | Henderson |
| turnip | radish | FALSE | 1060 | 39 | CMD |
| creature | gorilla | FALSE | 1558 | 543 | CMD |
| rabbit | squirrel | FALSE | 1260 | 249 | CMD |
| ship | battleship | FALSE | 1577 | 571 | CMD |
| giraffe | beast | TRUE | 1220 | 220 | Henderson |
| coyote | elephant | FALSE | 1017 | 28 | CMD |

Table 15: The top word pairs with maximum positive difference in ranks (CMD rank - Henderson rank). The rank is given out of 1635.

senses of the distributional estimate causes this behavior. One possibility to deal with this can be to take into account the similarity between word pairs in the ground metric. Overall, CMD does a good job at handling these pairs in comparison to the Henderson method.

| Hyponym candidate | Hypernym candidate | Ground Truth | CMD rank | Henderson rank | Better Method |
|---|---|---|---|---|---|
| box | mortality | FALSE | 116 | 1534 | Henderson |
| radio | device | TRUE | 110 | 1483 | CMD |
| television | system | TRUE | 5 | 1354 | CMD |
| elephant | hospital | FALSE | 52 | 1355 | Henderson |
| pistol | initiative | FALSE | 40 | 1316 | Henderson |
| library | construction | TRUE | 71 | 1335 | CMD |
| radio | system | TRUE | 6 | 1266 | CMD |
| bowl | artifact | TRUE | 223 | 1448 | CMD |
| oven | device | TRUE | 88 | 1279 | CMD |
| bear | creature | TRUE | 324 | 1513 | CMD |
| stove | device | TRUE | 167 | 1356 | CMD |
| saw | tool | TRUE | 461 | 1620 | CMD |
| television | equipment | TRUE | 104 | 1244 | CMD |
| library | site | TRUE | 87 | 1217 | CMD |
| battleship | bus | FALSE | 292 | 1418 | Henderson |
| pistol | device | TRUE | 70 | 1187 | CMD |
| battleship | vehicle | TRUE | 77 | 1175 | CMD |
| bowl | container | TRUE | 333 | 1431 | CMD |
| pub | construction | TRUE | 19 | 1116 | CMD |
| bowl | object | TRUE | 261 | 1334 | CMD |

Table 16: The top word pairs with maximum negative difference in ranks (CMD rank - Henderson rank). The rank is given out of 1635.

