# OpenReview forum: "Context Mover's Distance & Barycenters: Optimal transport of contexts for building representations"
_ICLR.cc/2019/Conference_

### Official Review · AnonReviewer2 · 2018-11-02
**The method is not very novel or not very well-motivated. The experiment results are interesting but mixed.**

**Rating:** 7
**Confidence:** 4

**Review:**


Pros:
I also study some related tasks and suspect that Wasserstein is helpful for measuring co-occurrence-based similarity. It is nice to see the effort in this direction.

Cons:
The methods are either not very novel or not very well-motivated. The experiment results are interesting but mixed. If the doubts about the experiments are clarified and the methods are motivated better (or the strengths/weaknesses are better analyzed), I will vote for acceptance.

Related work:
In addition to the work in the related work section, some other work also studied the NLP applications of Wasserstein, especially the ones (such as [1,2,3]) which are related to similarity measurement. The authors should include them in the related work section.

Question about experiments:
1. Why are the SIF scores reported in Table 1 much lower than the results reported in Arora et al., 2017 and in [4]?
2. If we compare CMD with DIVE + C * delta S, the proposed method wins in EVALution and Weeds, loses in Baroni, Kotlerman, BLESS, and Levy. If you compare DIVE + delta S (Chang et al. 2017) with DIVE + C * delta S, delta S also wins in EVALution and Weeds, loses in Baroni, BLESS, Kotlerman, and Levy (although CMD seems to be better than DIVE + delta S).
Based on the fact that your method has a high correlation with DIVE + delta S (Chang et al. 2017), I guess that CMD does not work very well when the dataset contains random negative samples, but work well when all the negative samples are similar to the target words. If my guess is right, the performance should be improved on average if you multiply the scores from CMD with the word similarity measurement.
3. To make it efficient, CMD seems to sacrifice some resolutions by using the K representative context. Does this step hurt the performance? Could you provide some performance comparison with different numbers of K to let readers know whether there is a tradeoff between accuracy and efficiency?
4. Since the results are mixed, I suppose readers would like to know when this method will perform better and the reasons for having worse results sometimes.

Writing and presentation suggestions/questions:
1. If the proposed method is a breakthrough, I am fine with the title but I think the experiment results tell us that Wasserstein is not all you need. I understand that everyone wants to have an eye-catching title for their paper. The title of this paper indeed serves this purpose. Since the strategy is effective, more and more people might start to write papers with a title like this. However, having lots of paper called "XXX is all you need?" or "Is XXX all you need?" is definitely not good for the whole community. Please use a more specific title such as Context Mover Distance or something like that.
2. The last point in the contribution is not supported by experiments. I suggest that the authors move this point to the future work section.
3. It is good to see some negative results like Baroni in Table 2. Results on other datasets should not be put into Table 4 in Appendix.
4. Using Wasserstein barycenter to measure sentence similarity seems to be novel, but the motivation is not very clear. Based on A.6, we could see that for each sentence, authors basically find the representative word which is most likely to co-occur with every word in the sentence (has the highest average relatedness rather than similarity) and measure the Wasserstein distance between the co-occurrence probability distribution. I suppose sometimes relatedness is a better metric when measuring sentence similarity, but I think authors should provide some motivative sentence pairs to explain when that is the case.
Using Wasserstein to detect hypernym seems to also be novel, but the motivation is also not clear. Again, a good example would be very helpful.
This point is also related to the last question for experiments.

Minor writing suggestions:
1. In section 3, present the full name of CITE
2. If you put some important equations to the appendix (e.g., the definition of SPPMI_{alpha,gamma}), remember to point readers to the appendix.
3. In the second paragraph of section 7, Nickel & Kiela, 2017 is a method supervised by a hierarchical structure like WordNet rather than a count-based or word embedding based methods.
4. In Chang et al., the training dataset is not Wikipedia dump from 2015. This difference of evaluation setup should be mentioned somewhere (e.g., in the caption of Table 2).
5. The reference section is not very organized. For example, the first name of Benotto is missing for the PhD thesis "Distributional Models for Semantic Relations: A Study on Hyponymy and Antonymy". The arXiv papers are cited using different formats. Only some papers have URL. The venue's names are sometimes not capitalized. Gaussian embedding is cited twice, etc.


[1] Kusner, M., Sun, Y., Kolkin, N., & Weinberger, K. (2015). From word embeddings to document distances. In International Conference on Machine Learning (pp. 957-966).
[2] Xu, H., Wang, W., Liu, W., & Carin, L. (2018). Distilled Wasserstein Learning for Word Embedding and Topic Modeling. NIPS
[3] Rolet, A., Cuturi, M., & Peyré, G. (2016, May). Fast dictionary learning with a smoothed wasserstein loss. In Artificial Intelligence and Statistics (pp. 630-638).
[4] Perone, C. S., Silveira, R., & Paula, T. S. (2018). Evaluation of sentence embeddings in downstream and linguistic probing tasks. arXiv preprint arXiv:1806.06259.

---

> ### Author Response · Authors · 2018-11-20
> **Thanks a lot for the detailed feedback and suggestions. Please find our response to all comments: (1/2)**
>
> These suggestions have been quite helpful and certainly helped us further improve the paper. Thank you.
>
> |  Related Work & Novelty:
>
> Thanks for sharing these articles which are indeed interesting applications of Wasserstein distance in NLP. The Word Mover’s distance paper from Kusner, et.al. (2015) has already been discussed more than once. We have included [Rolet, et. al, 2016] and the concurrent work by [Xu, et. al. 2018] in our revised related work as it might be fruitful for the readers to know.
>
> But, since the focus of all these works is on transporting the words (as points) directly to form a metric over documents, an important issue still remains. This is the inability to define a suitable metric for comparing words, phrases or sentences which lie below documents in the levels of grammatical hierarchy. When we define the transport over contexts, it not only handles these lower levels of hierarchy, but also offers the ability to go up the hierarchy with the Wasserstein barycenter.
>
> Thus, we are confident that our formulation of Context Mover's Distance & Barycenters is novel and opens the floor to further possibilities.
>
> | Questions about experiments:
>
> | Difference in SIF scores
>
> The SIF scores in [Arora, et. al.] and in [4] are based on vector embeddings trained on CommonCrawl (840 Billion tokens), while the results in our paper are based on embeddings trained on the Toronto Book Corpus (~ 0.98 Billion tokens) and thus the difference. The CommonCrawl vectors are publicly available but the co-occurrence information isn’t, which is required for our method. Hence, we use Toronto book corpus for a fair and accurate comparison of SIF and our method.
>
> | if you multiply the scores from CMD with the word similarity
>
> Thanks for pointing this, and your guess seems probable to help in further improving the performance.  Note that our focus over here was to compare CMD between the words (using Henderson embeddings for the ground metric) versus just using these embeddings alone.  There are probably several nice tricks (as you suggested) that can be further used along with CMD to improve the performance. This although wouldn’t have helped us much to validate our hypothesis of using distributional estimates over just point embeddings.
>
> | Could you provide some performance comparison with different numbers of K
>
> Yes, you are right that it would be useful for the readers to know about the trade-off. We have taken this into account in our revision, where in Section A.8 of the Appendix, we plot the performance versus number of clusters for the three best variants in Table 1 as well as for the average across these variants.
>
> We observe (c.f. Fig 3) that on average increasing the number of clusters until around K=300, the performance significantly improves. Beyond that it mostly plateaus (± 0.5) and isn't worth the increased computation time. Please check out section A.8 for more details.

---

> ### Author Response · Authors · 2018-11-20
> **Thanks a lot for the detailed feedback and suggestions. Please find our response to all comments: (2/2)**
>
> (continued from Part1 )
>
> | presentation suggestions
>
> | Title
>
> We agree with your statement in general. But, we would first like to clarify that through our previous title, we were not claiming that Wasserstein is or isn’t all you need. Rather our motivation was to show how the tools of Wasserstein distance and barycenter which are at the heart of our framework can hold significance in problems with a co-occurrence structure (like your remark that Wasserstein could be “helpful for measuring co-occurrence-based similarity”).
>
> Natural language is one such domain with an inherent co-occurrence structure and we show that our framework is competitive (and can also beat state-of-the-art) on sentence representation/similarity and hypernymy, which are quite independent in their nature. All this without requiring additional training.
>
> You are completely right about how such naming practice isn’t useful for the community. Taking this into account: we have changed the title to “Context Mover’s Distance & Barycenter: Optimal transport of contexts for building representations”, in order to make it more factual and reflective about our method.
>
> | Last point in contribution:
>
> Yes, thanks for this tip. We have written it as a future work direction.
>
> | basically find the representative word which is most likely to co-occur with every word in the sentence
>
> Yes, this is right to an extent. In the qualitative analysis, we wanted to understand the nature of resulting wasserstein barycenter of a sentence and hence looked at the nearest words in the vocabulary. But while computing similarity between sentences, we do it by directly measuring the CMD between the produced barycenters of sentences and not between the representative word.
>
> Most probably looking at the nearest neighbors in the space of sentences would give a more finer picture, but such a space is huge and looking amidst just a few sentences would be biased.
>
> | Minor writing suggestions
>
> Thanks you so much for pointing these out. We admit that the reference section was a bit untidy and have organized it better. All other suggestions have been taken into account as well and can be seen in the revision.
>
> Hope this addresses all of your questions about the experiments. In light of this response, it would be great if you can reconsider the score for your review. And, please feel free to ask about any questions or parts which you might need clarification.

---

> > ### Comment · AnonReviewer2 · 2018-11-21
> > **The authors' revision is good, but the motivations of Barycenters and hypernym experiments are still not very clear**
> >
> > Thanks for the revision and doing the additional analysis about cluster number k. The modification and clarification move my position to borderline (neither accept nor reject).
> >
> > Nevertheless, most of my concerns about the motivation remain unsolved. The paper only tries to justify using Wasserstein distance between two words (e.g., in Figure 1). If the experiments are done in some word similarity benchmarks, the justification is fine. However, the experiment results in this paper are about measuring sentence similarity and detecting hypernyms, but explanations of why and when this could work well in these problems are not sufficient.
> >
> > The experimental results are mixed and the method is complicated from practical perspectives (i.e., measuring Wasserstein distance based on co-occurrence in an efficient way is not that easy to be implemented). This will prevent other researchers from trying the proposed method unless they have some reasons/intuitions to believe the proposed method would have a high chance to work better in their application of interest. We need some more analysis to tell us which kinds of corpora we should try the proposed method in. One of the simple things authors could do is to list some similar sentence pairs or word pairs with a hypernym relation which are ranked higher using the proposed method than using the baselines and try to explain why the proposed method works better.
> >
> > I understand that some biases might be created during the analysis as you said, but some methods could be used to reduce the biases. For example, you could also list some examples where the baselines outperform the proposed method. When the results are mixed, I believe more analysis is unavoidable. Otherwise, people do not even know what are the lessons experiments tell us and what are problems this paper actually solves.

---

> > > ### Author Response · Authors · 2018-11-27
> > > **Thanks for the suggestions, we have added the qualitative analysis and can be found in sections B and C.**
> > >
> > > Thank you so much for your prompt and helpful advice about carrying the qualitative analysis. As noted in the general comment above, we have added Sections B and C in the appendix about our performed qualitative analysis for the case of sentence similarity as well as hypernymy.
> > >
> > > Sentence similarity: The details of our evaluation procedure can be found in section B.1. The analysis is carried out three datasets: STS14 News, STS15 Images and STS 14 WordNet. We also discuss our observations from the listed examples and mention explanations about where and why our proposed method works. For a quick overview, we suggest looking at observations in sections B.2 (from STS14 News) and some conclusions in B.3. We also encourage you to look at similar observations derived from analysing the other two datasets in Section B.5.
> > >
> > > In section B.4, we also present some observations about the effect of sentence length on both methods, and in turn comment briefly about the kind of corpora.
> > >
> > > Hypernymy: We have performed a qualitative analysis here also and results can be found in Section C.2 (in both cases of maximum positive and negative difference in ranks).
> > > Btw, we would like to remark that in Figure 1, we had used the entailment ground metric, but considered histograms with just some top contexts for illustration purpose. We have added this information in the Fig. 1 caption.
> > >
> > > We believe that the observations made through these qualitative analysis experiments, should clarify questions regarding the motivation. In particular, the observations and conclusions mentioned in these examples should give intuitions about when and why our method would have a better chance to work. We have also noted points about some lessons/ideas for future  (c.f. comments in these section, like about ground metric and complementary nature of errors).
> > >
> > > Finally, we would like to emphasize that our code to obtain histograms, CMD & CoMB as well as pre-built histograms would be made available (which would avoid the need to carry out co-occurrence related computation). Plus, a package would be released for making the standard evaluations on Hypernymy in a simple manner.
> > >
> > > In the end, thanks a lot for encouraging us to do this analysis. This has not only re-affirmed our past beliefs, but also given us some more practical insights. We hope this improves your stance about the paper and please let us know if you have any other comments or suggestions. Thanks!

---

> > > > ### Comment · AnonReviewer2 · 2018-11-27
> > > > **I have moved my position to accept**
> > > >
> > > > The paper proposed two novel unsupervised methods, applying Barycenters to measure sentence similarity and applying Wasserstein distance to detect hypernym.
> > > > The results show that the proposed method is efficient, outperforms SIF in many cases without accessing global statistics, and also outperforms DIVE + dS, which is the state of the art in many unsupervised hypernym detection tasks.
> > > > I believe these results are promising and are sufficient to demonstrate the usefulness of this proposed methods.
> > > >
> > > > Final suggestions:
> > > > 1. Your qualitative analysis does not tightly connect to your methods. You only say in many situations, the proposed method might be better than SIF, but you do not explain why using Barycenters will lead to that. Analyzing why could really provide the motivations of your method.
> > > > 2. The results show that the method significantly outperforms DIVE + dS, but the result is still mixed compared with DIVE + C*dS. I encourage authors to multiply your score with some word similarity measurements (like DIVE + W*dS in Chang et al. 2017) and really show that the proposed method outperforms DIVE + W*dS on average.

---

> > > > > ### Author Response · Authors · 2018-11-29
> > > > > **Thank you for your consideration and we will be working on the suggestions**
> > > > >
> > > > > Also, thank you for spending the time and valuable feedback.
> > > > >
> > > > > If you look at the weighted averaging part in SIF (ignore the principal component part for a moment), it can be thought of as a special case of taking the Wasserstein barycenters with the histograms as Diracs at the word. In our case we have a histogram over the contexts of a word which inherently contains richer information than just a Dirac, and hence that's why we believe doing Barycenter is better than SIF in several cases.
> > > > >
> > > > > Though your suggestion is also right, in the sense that our current qualitative analysis mostly shows examples to give some understanding/intuition, and we agree with you that further analysis to demonstrate these points can be quite helpful. This unfortunately needed more time than available in the rebuttal session, but we aim to work on this by the camera ready deadline.
> > > > >
> > > > > The second suggestion is very relevant too. In fact for sentence similarity task as well (like examples where the difference between sentences stems from just the subject/object), we think such a related modification in the ground metric can turn out to be useful. We will carry out experiments on this in the meanwhile and include the obtained results once the revision system is open again.
> > > > >
> > > > > We really appreciate your detailed feedback and suggestions, which has helped in improving the paper. Please let us know if you have further questions/suggestions.

---

### Official Review · AnonReviewer1 · 2018-11-02
**Interesting approach, proposes representation augmentation as opposed to representation learning and the proposed distance not used for training.**

**Rating:** 6
**Confidence:** 4

**Review:**

The paper proposes a method to augment representation of an entity (such as a word) from standard "point in a vector space" to a histogram with bins located at some points in that vector space. In this model, the bins correspond the context objects, the location of which are the standard point embedding of those objects, and the histogram weights correspond to the strength of the contextual association. The distance between two representations is then measured with, Context Mover Distance, based on the theory of optimal transport, which is suitable for computing the discrepancy between distributions.
The representation of a sentence is proposed to be computed as the barycenter of the representation of words inside.
Empirical study evaluate the method in a number of semantic textual similarity and hypernymy detection tasks.

The topic is important. The paper is well written and well structured and clear. The method could be interesting for the community. However, there are a number of conceptual issues that make the design a little surprising. First, the method does not learn the representations. Instead, augments a given one and computes the context mover distance on top of that. But, if the proposed context mover distance is an effective distance, maybe representations are better to be "learned" based on the same distance rather than being received as inputs.
Also, whether an object is represented as a single point or as a distribution seems to be an orthogonal matter to whether the context predicts the entity or vice versa. This two topics are kind of mixed up in the discussions in this paper.

Other issues:

- One important technicality which seems to be missing is the exact value of p in Wp which is used. This becomes important for barycenters computations and the uniqueness of barycenters.
- Competitors in Table 1 are limited. Standard embedding methods are missing from the list.
- Authors raise a question in the title of the paper, but the content of the paper is not much in the direction of trying to answer the question.
- It is not clear why the "context" of hyponym is expected to be a subset of the context of the hypernym. This should not always be true.
- Table 4 gives the impression that parameter might not be set based on performance on validation set, but instead based on the performance on the test set.

- Minor:
of of
data ,
by
byMuzellec
CITE

Overall, comparing strengths and shortcomings of the paper, I vote for the paper to be marginally accepted.

---

> ### Author Response · Authors · 2018-11-20
> **Thanks for the very encouraging and helpful feedback. Response to all the questions/suggestions below:**
>
> Also, we appreciate that you recognize the importance of topic and find the paper to be well structured and clear.
>
> | First, the method does not learn the representations. Instead, augments a given one and computes the context mover distance on top
>
> This is a great point. Under the proposed framework, our first aim was to investigate the out-of-box performance, i.e.,  by just using off-the-shelf embeddings like GloVe to form the ground metric. We realized through the empirical experiments that this already served as a decent starting point, where we were able to perform competitively on sentence similarity and beat state-of-the-art methods in hypernymy on several datasets.
>
> | maybe representations are better to be "learned"
>
> We totally agree with your remark, and an excellent direction to pursue in the future would be to learn the representations based on the Context Mover's Distance and Barycenter. An example of such a learning procedure could be like CBOW with negative sampling. Another would be to learn the ground metric based on supervised tasks.
>
> Overall, our paper takes the first step and shows that the framework holds promise via  'augmenting' the representations, and provides for an interesting direction to explore with the learning aspect.
>
> | whether an object is represented as a single point or as a distribution
>
> A given object (e.g. 'cat')  which we seek to represent is considered to be as a distribution/histogram over contexts (e.g. {'milk', 'drinking', 'cute', …}). So, over here the distribution is over all possible contexts with which the object can co-occur. Hence, it suffices to denote each individual context (like 'cute') as a single point here.
>
> | Other issues:
>
> | the exact value of p
>
> This is an important question as  (Agueh and Carlier, 2011) have shown that when the underlying space is Euclidean and p=2, there exists a unique minimizer to the Wasserstein barycenter problem. Since we are anyways solving the regularized Wasserstein barycenter (Cuturi & Doucet, 2014) over here instead of the exact one, the particular value of p is less of an issue. Empirically in the sentence similarity experiments, we observed p=1 to perform better than p=2 (by ~ 2-3 points). We have included this in our revision.
>
> | Standard embedding methods are missing from the list.
>
> We would like to remark that our emphasis is on unsupervised sentence representation methods. Including other standard methods such as InferSent (Conneau, et.al, 2017) or USE (Cer, et.al, 2018) would lead to an unfair comparison as they are supervised or based on conversational data respectively.  As a reference, we have included scores for Skip-Thought, the unsupervised method from (Kiros et al., 2015), in our revision.  Overall, our focus is to compare unsupervised methods which can just build up on the word-embedding information (i.e., training-free methods).
>
> |  Authors raise a question in the title
>
> At the heart of our method are the core tools from Wasserstein geometry, which are then demonstrated on two important tasks in NLP: sentence representation & similarity, and hypernymy detection. Hence, our intention is to emphasize how helpful these tools can be and thus we raise the question.  In fact, we have decided to change the title to “Context Mover’s Distance & Barycenters: Optimal transport of contexts for building representations”, in order to make it more factual and reflective.
>
> | It is not clear why the "context" of hyponym is expected to be a subset of the context of the hypernym.
>
> Thanks for raising this point. It is indeed true that the context of the hyponym may not always be a subset of the context of the hypernym. The essential idea originates from the Distributional Inclusion Hypothesis (Geffet & Dagan, 2005), which states that a word ‘v’ entails another word ‘w’, if “the characteristic contexts of v are expected to be included within all w's contexts (but not necessarily amongst the most characteristic ones for w).”
>
> However, in contrast to the above distributional inclusion hypothesis, we see our method as a relaxation of this strict condition, by having an entailment based ground cost between the contexts while using CMD.
>
> | the impression that parameter might not be set based on performance on validation set
>
>  In the hypernymy experiments, only exploring the limited space of hyper-parameters (< 10 configurations) listed in Table:4 were sufficient to obtain these results. Hence, you are correct to notice that the parameters are not set based on performance on validation set and rather the benchmarks themselves. As a matter of fact, we are currently experimenting on a much larger space of hyper-parameters with a validation set in place and we will be updating our revision soon.
>
> | Minor
>
> Thanks for pointing them, have been corrected.
>
> We hope this clarifies the questions you had about the shortcomings and in general. Please feel free to post any other comments that you might have! :)

---

> ### Author Response · Authors · 2018-12-03
> **Feedback on revision and response**
>
> Dear Reviewer 1,
>
> We were wondering if you have any additional suggestions/comments on our revision and response? In particular, we've included detailed qualitative analysis for sentence similarity and hypernymy (Sec B & C), results with validation (in Table 2) for hypernymy detection, clarification about STS baselines, and answers to individual questions.
>
> We greatly appreciate the time that you've given to the paper so far and for providing the constructive feedback. Thank you so much!

---

### Official Review · AnonReviewer3 · 2018-11-04
**Interesting method, but needs more to show it is useful**

**Rating:** 4
**Confidence:** 4

**Review:**

The submission explores a new form of word representation based on a histogram over context word vectors, allowing them to measure distances between words in terms of optimal transport between these histograms. The authors speculate that this may allow better representations of polysemous words. The approach is mathematically elegant, and requires no additional training on top of existing approaches like Glove. To improve efficiency, they use clustering on context vectors. They present results on various semantic textual similarity and hypernym detection tasks, outperforming some baselines.

The paper presents itself as an alternative to word embeddings as a way of representing words. As far as I can tell, their method only really allows a way of computing distances between pairs of word representations, which hasn't been a useful concept for the vast majority of cases that word embeddings have been used for (translation, QA, etc.). Point estimates are at least very convenient to work with. That doesn't mean the proposed approach is useless, but the paper needs to give a much stronger motivation for when and why measuring distances between words may be helpful.

The experiments are a bit underwhelming. STS and hypernymy detection are somewhat unimpressive tasks to work - I'm not aware of any results on these tasks that have generalized to more realistic applications like translation or question answering. I think for publication with just these tasks, the method would need to show a dramatic breakthrough, which the submission definitely does not. The STS baselines are very simple bag-of-words approaches, and even then the results for the SIF baseline are much lower than those reported by Arora et al. (2017). At the least, there should be a comparison with the current state of the art. On the hypernymy task, what validation data was used? Unfortunately I'm not able to suggest better experiments, because I can't think of cases where their method would be useful.

The paper is significantly weakened by frequently making very strong claims based on rather limited experimental results (for one example, "we illustrate how our framework can be of significant benefit for a wide variety of important tasks" feels like quite a stretch). It would be much improved if some of the language was toned down.

Overall, the paper introduces a mathematically elegant method for representing words as distributions over contexts, and for computing distances between these words. For acceptance, I think the paper needs to better motivate why the method could be useful, and back that up with more convincing experiments.

---

> ### Author Response · Authors · 2018-11-21
> **Thanks for the feedback. Response to questions/criticisms below (1/2):**
>
> Thank you so much for your detailed feedback and taking the time. We are happy to know that the reviewer acknowledges the elegance of the approach and that it doesn’t require additional training on top of existing approaches.  Please find the answer to the specific questions as follows:
>
> | only really allows a way of computing distances between pairs of word representations
>
> We respectfully disagree with this statement. Representing words as distribution over contexts and then using Context Mover's Distance (CMD) is just half of the picture.
>
> - In addition, our method importantly provides a principled manner to extend this to form representation of sentences via Wasserstein barycenters.
>
> - Since this obtained representation is also a distribution over contexts, we can again employ CMD to compare the semantics of two sentences.
>
> - Further, as the Context Mover's Distance and Barycenters are parameterized by the ground metric, this offers the flexibility to compare words and sentences with respect to different cost such as that of entailment.
>
> | question on use-cases
>
> We agree that just computing distance between word representations may not have been as useful in translation, QA. But, since our contribution is beyond just word representations, consider the following cases:
>
> - Using CMD for downstream tasks like QA or NLI: Currently, we rely on existing approaches like GloVe to form the ground metric between words. But, for a supervised task like QA/NLI, it would be more effective to learn this ground metric. For example, one can learn a linear (or non-linear) projection M that takes in the GloVe embeddings and maps it to a space that better measures the cost of transport between two points as required for the downstream task. Thus the objective/loss can be designed so that the hypothesis and the true premise are closer in CMD, while the hypothesis and the false promised are far away with respect to CMD.
>
> - Somewhat similar to the above point, Tay et.al, 2017 [1] have explored the idea of using the Hyperbolic representation space for QA and this seems to have a strong performance [2]. Of course this doesn't imply that doing it in the Wasserstein space would be better or worse, but points to the potential of utilizing different representation spaces for tasks like QA.
>
> - For low-resource languages, Qi et.al, 2018 [3], demonstrate the effectiveness of having pre-trained word representations for translation, and thus having improved representations of words for instance can be useful in such a scenario.
>
> [1] Tay et.al, 2017: "Hyperbolic Representation Learning for Fast and Efficient Neural Question Answering", https://arxiv.org/pdf/1707.07847.pdf
> [2] https://aclweb.org/aclwiki/Question_Answering_(State_of_the_art)
> [3] Qi et. al, 2018: "When and Why are Pre-trained Word Embeddings Useful for Neural Machine Translation?", NAACL, http://www.aclweb.org/anthology/N18-2084
>
> | point estimates easy to work with
>
> We agree with you that point estimates are easier to work. The focus on distributional estimates is comparatively recent, mainly originating from Vilnis & McCallum, 2014. This doesn't imply that we should stop focussing on such distributional estimates, but rather strengthen our relatively weak toolkit for dealing with them, given the potential offered by representing entities as distributions.
>
> Motivated by the results of our proposed approach, an exciting future exploration would be to try searching for an embedding space where euclidean distance mimics Wasserstein distance (Courty, et.al, 2017) for the particular formulation of Context mover's distance.
>
> | Motivate : why the method can be useful
>
> The motivation for representing words and sentences as distributions over the contexts is that it inherently allows the ability to capture the various senses under which they can appear. For instance, the Table 3 in Appendix A.6 tries to qualitatively probe the sentence representation produced with Wasserstein barycenter, by looking at the nearest neighbor in the vocabulary (with respect to CMD). Here, the method seems to capture a varied range of contexts, like for the ambiguous sentence “he lives in europe for”, the obtained closest neighbors include: ‘decades’, ‘masters’, ‘majority’, ‘commerce'.  More such examples can be found in this section A.6, and we hope these examples address your questions about motivation.

---

> ### Author Response · Authors · 2018-11-21
> **Thanks for the feedback. Response to questions/criticisms below (2/2):**
>
> (Continued from Part1)
>
> | Experiments
>
> |  results for the SIF baseline are much lower
>
> The SIF scores in Arora, et. al., are based on vector embeddings trained on CommonCrawl (840 Billion tokens), while the results in our paper are based on embeddings trained on the Toronto Book Corpus (~ 0.98 Billion tokens) and thus the difference. The CommonCrawl vectors are publicly available but the co-occurrence information isn’t, which is required for our method. Hence, we use Toronto book corpus for a fair and accurate comparison of SIF and our method.
>
> | there should be a comparison with the current state of the art
>
> We would like to remark that our emphasis is on unsupervised sentence representation methods. Including other standard methods such as InferSent (Conneau, et.al, 2017) or USE (Cer, et.al, 2018) would lead to an unfair comparison as they are supervised or based on conversational data respectively.  As a reference, we have included scores for Skip-Thought, the unsupervised method from (Kiros et al., 2015), in our revision.  Overall, our focus is to compare unsupervised methods which can just build up on the word-embedding information (i.e., training-free methods).
>
> | on the hypernymy task, what validation data was used
>
> In the hypernymy experiments, only exploring the limited space of hyper-parameters (< 10 configurations) listed in Table:4 were sufficient to obtain these results. Hence, you are correct to notice that the parameters are not set based on performance on validation set and rather the benchmarks themselves. As a matter of fact, we are currently experimenting on a much larger space of hyper-parameters with a validation set in place and we will be updating our revision soon.
>
> | writing and language style
>
> Thanks for the suggestions about the writing style, we have taken them into account.
>
> By addressing these criticisms, we hope that it changes your viewpoint of the paper, and that you re-evaluate the score for the review.  Lastly, please let us know if you have any other comments or questions in mind.

---

> ### Author Response · Authors · 2018-12-03
> **Feedback on revision and response**
>
> Dear Reviewer 3,
>
> We greatly appreciate the time that you've given to the paper so far and for providing the feedback. We were wondering if you have any additional suggestions/comments on our revision and response? In particular, we've included detailed qualitative analysis for sentence similarity and hypernymy (Sec B & C), results with validation (in Table 2) for hypernymy detection, clarification about STS baselines, and answers to individual questions.
>
> Overall, please don’t hesitate to let us know if there are any additional clarifications that we can provide, as we would love to convince you of the merits of this work. Thank you so much!

---

### Author Response · Authors · 2018-11-21
**Summary of Revision: 20th Nov, 2018**

We would like to thank the reviewers for all their great suggestions which have definitely have helped shape our paper better. Following which we have:

- Added results of experiments analysing the effect of number of clusters on performance. The results and the plot can be found in Section A.9 of the Appendix.
- Explained the difference of SIF results in the paper and included the baseline of Skip-thoughts with results taken from [Arora, et.al, 2017].
- Changed the title to the more factual “Context Mover's Distance & Barycenters: Optimal transport of contexts for building representations”.
- Addressed specific questions raised by the reviewers.
- Made the writing more clear in the required places and added missing references.

In addition, we have added a sub-section about “Online computation” in the Sentence similarity experiments to highlight a particular aspect of SIF that has remained under the rug.

Specifically, the principal component removal is carried out (in a topic-wise manner) on the sentence embeddings in the test set. This gives them an advantage as it utilizes the inter-sentence information in the test set, but even without utilizing such information our methods perform competitively and leads to an overall gain.

We greatly acknowledge the helpful suggestions from the reviewers. Also, we welcome any other comment/questions that the reviewer or the area chairs or the public might have. Thank you so much!

---

### Author Response · Authors · 2018-11-27
**Summary of revision: 26th Nov, 2018:**

We again thank the reviewers for their fruitful comments and suggestions. This revision contains the following:

- Updated results for Hypernymy when validation is performed on HypNet train set, following the procedure in DIVE (Chang et.al., 2018). The scores are in the same ballpark of what was previously presented.
- Added a new Section B in the appendix on qualitative analysis of sentence similarity for 3 datasets, giving some insights about which methods work when.
- Included some observations about the effect of sentence length in Section B.4.
- Also, added a Section C in appendix, showing the results of a similar qualitative analysis in the case of hypernymy detection.

Overall, we hope that these additional experiments about the qualitative analysis should clarify concerns about the motivations of using context mover's distance & barycenters and where it can be helpful. Thus, we kindly request the reviewers to reassess their evaluations about the same.

Lastly, we are happy to address any other further queries which the reviewers might have. Thank you for your time and feedback!

---

### Author Response · Authors · 2018-12-14
**Code release (package for hypernymy evaluation)**

Dear reviewers and area chair,

As promised earlier, we have released a python package for carrying out Hypernymy evaluations in an easy manner and with all the datasets organized in one place. The link is https://github.com/context-mover/HypEval

We also aim to release other parts of the code soon on the same GitHub profile: https://github.com/context-mover . Thanks for your time and feedback.

---

### Meta-Review · Area_Chair1 · 2018-12-14
**Specific tasks and mixt results in empirical analysis limit significance**

**Confidence:** 3
**Recommendation:** Reject

**Metareview:**

The paper proposes to build word representation based on a histogram over context word vectors, allowing them to measure distances between words in terms of optimal transport between these histograms. An empirical analysis shows that the proposed approach is competitive with others on semantic textual similarity and hypernym detection tasks. While the idea is definitely interesting, the paper would be streghten by a more extensive empirical analysis.